# Endocytic vesicles act as vehicles for glucose uptake in response to growth factor stimulation

Ryouhei Tsutsumi ●[1,2,3] ✉, Beatrix Ueberheide[3,4,5,6], Feng-Xia Liang ●[3,7], Benjamin G. Neel[3], Ryuichi Sakai ●[1] & Yoshiro Saito ●[2]

Glycolysis is a fundamental cellular process, yet its regulatory mechanisms remain incompletely understood. Here, we show that a subset of glucose transporter 1 (GLUT1/SLC2A1) co-endocytoses with platelet-derived growth factor (PDGF) receptor (PDGFR) upon PDGF-stimulation. Furthermore, multiple glycolytic enzymes localize to these endocytosed PDGFR/GLUT1-containing vesicles adjacent to mitochondria. Contrary to current models, which emphasize the importance of glucose transporters on the cell surface, we find that PDGF-stimulated glucose uptake depends on receptor/transporter endocytosis. Our results suggest that growth factors generate glucose-loaded endocytic vesicles that deliver glucose to the glycolytic machinery in proximity to mitochondria, and argue for a new layer of regulation for glycolytic control governed by cellular membrane dynamics.

High rates of glycolysis correlate with cell stemness and cancer[1,2], yet the underlying mechanisms are incompletely understood. The rate-determining steps of glycolysis include uptake of extracellular glucose by transporters and subsequent phosphorylation by hexokinases (HKs) to prevent re-release of glucose to the extracellular space. Among the mammalian facilitated glucose transporter (GLUT) family members SLC2A1-14, the ubiquitously expressed GLUT1 (SLC2A1), the renal tubular cell-, hepatocyte- and pancreatic β cell-specific GLUT2 (SLC2A2), the neuron-and placenta-specific GLUT3 (SLC2A3), and the adipose tissue- and striated muscle-specific GLUT4 (SLC2A4) are well characterized[3]. GLUT4 is the only glucose transporter known to dynamically increase glucose uptake without the need for new transcription/translation; in response to insulin, it translocates from intracellular structures to the plasma membrane[3,4]. However, earlier work showed that stimulation of fibroblasts, which do not express GLUT4, with epidermal growth factor (EGF) resulted in elevation of glucose uptake within 15 min[5]. This observation cannot be explained by current models for glycolysis regulation in such cells, which feature transcriptional upregulation of glucose transporters and/or glycolytic enzymes[1]. One study reported that fibroblasts treated with 12-O-tetradecanoylphorbol-13-acetate (TPA) activate GLUT1 via PKC-dependent phosphorylation on S226[6], although whether PKC-dependent phosphorylation is fully responsible for promoting growth factor-evoked glucose uptake has remained unclear.

Growth factors bind and activate their cognate receptor tyrosine kinases (RTKs) at the plasma membrane, resulting in protein tyrosyl phosphorylation-dependent signaling and receptor endocytosis[7,8]. Endocytosis results in RTK inactivation/degradation unless the RTK is recycled to the plasma membrane. However, multiple studies also implicate intracellular vesicles containing active RTKs as signaling platforms[9,10]. We developed a method, analogous to one reported previously[11], that enables efficient and quick recovery of growth factor-evoked, RTK-containing endocytic vesicles. Here, we report that upon growth factor stimulation, GLUT1 and glycolytic enzymes co-localize

[1]Kitasato University School of Medicine, Sagamihara 252-0374, Kanagawa, Japan. [2]Graduate School of Pharmaceutical Sciences, Tohoku University, Sendai 980-8578, Miyagi, Japan. [3]Perlmutter Cancer Center, NYU Grossman School of Medicine, NYU Langone Health, New York, NY 10016, USA. [4]Proteomics Laboratory, NYU Grossman School of Medicine, NYU Langone Health, New York, NY 10016, USA. [5]Department of Biochemistry and Molecular Pharmacology, NYU Grossman School of Medicine, NYU Langone Health, New York, NY 10016, USA. [6]Department of Neurology, NYU Grossman School of Medicine, NYU Langone Health, New York, NY 10016, USA. [7]Microscopy Laboratory, NYU Grossman School of Medicine, NYU Langone Health, New York, NY 10016, USA. ✉e-mail: tsutsumi.ryohei@kitasato-u.ac.jp

with such vesicles. Furthermore, our data suggest that receptor/GLUT1 co-endocytosis serves to deliver glucose to glycolytic enzymes in proximity to mitochondria, allowing rapid adaptation of cellular glucose metabolism to mitogenic signals.

## Results

### Nanoparticle-based isolation identified PDGFR-containing endocytic vesicle-associated proteins

PDGFR has α and β isoforms, which homo- or heterodimerize in response to different dimeric PDGF isoforms (PDGF-AA, PDGF-BB, PDGF-AB, PDGF-CC, PDGF-DD)[12]. Serum-starved Swiss 3T3 mouse fibroblasts, which express PDGFRα and β, were treated for 5 min with biotinylated PDGF-BB bound to streptavidin-coated magnetic iron oxide nanoparticles with a diameter of 10 nm. These particles bind to surface PDGFRs and, following PDGF stimulation, are transported with the receptor into endocytic vesicles (~100 nm) (Supplementary Fig. 1a, b). Nanoparticles remaining on the cell surface were stripped by acid washes, and post-nuclear supernatants were passed through a magnetic separation column to enrich for nanoparticle-containing structures (Fig. 1a), resolved by SDS-PAGE, and silver-stained (Fig. 1b). Treating cells with the PDGF-BB-conjugated nanoparticles at 4 °C to inhibit receptor endocytosis, or treatment with nanoparticles and unconjugated PDGF-BB at 37 °C, resulted in recovery of significantly fewer proteins (Fig. 1b, Supplementary Fig. 1c).

Liquid chromatography-tandem mass spectrometric (LC-MS/MS) analysis of isolates from cells stimulated with PDGF-BB plus unconjugated nanoparticles (control) or with PDGF-BB-conjugated nanoparticles (endocytic vesicle fraction) identified 2781 proteins at a threshold of 2 unique peptides per protein. Of these, 1541 were unique to the endocytic vesicle fraction (Fig. 1c, Supplementary Data 1) and were enriched for proteins annotated as having endocytosis- or vesicular transport-related functions (Fig. 1d), supporting the validity of our method. The proteins detected in response to either type of stimulation could reflect ligand-stimulated bulk endocytic transport of nanoparticles into cells as reported previously[13]. Even so, we detected more of these proteins, especially endocytosis-related proteins, in the PDGF-BB-conjugated nanoparticle fraction than in the unconjugated PDGF-BB plus nanoparticle control (Supplementary Fig. 1d). Immunoblotting experiments confirmed that PDGFR and endocytosis-related proteins, including clathrin heavy chain (CHC), dynamin 2, and early endosome antigen 1 (EEA1), were detected almost exclusively in the endocytic vesicle fraction (Fig. 1e). The endocytic vesicle fraction also contained proteins related to vesicle maturation and transportation, as well as cell signaling (Supplementary Data 1, Supplementary Fig. 1e, f), comporting with the notion that these vesicles function as signaling platforms[9,10]. In addition, this fraction contained Golgi apparatus, endoplasmic reticulum (ER) and mitochondrial proteins (Supplementary Fig. 1e, Supplementary Data 1). These could represent organellar contamination but also might indicate recovery of organelle-endocytic vesicle contacts.

### PDGFR co-endocytoses with GLUT1 and colocalizes with most glycolytic enzymes

Unexpectedly, GLUT1, as well as most glycolytic enzymes, including hexokinase (HK1 and HK2), glucose-6-phosphate isomerase (GPI), phosphofructokinase (PFKL), aldolase (ALDOA), triosephosphate isomerase (TPI1), glyceraldehyde 3-phosphate dehydrogenase (GAPDH), phosphoglycerate kinase (PGK1), phosphoglycerate mutase (PGAM1), enolase (ENO1/2), and pyruvate kinase (PKM1/2, PKLR), also were enriched in the endocytic vesicle fraction by MS (Fig. 1f) and by immunoblotting (Fig. 1g). Validations of the antibodies against glycolytic enzymes used in the study are shown in Supplementary Fig. 2. In parallel (and to rule out their artifactual recovery owing to organellar contamination of the vesicle fraction), we monitored the subcellular localization of these proteins by confocal immunofluorescence

microscopy. In PDGF-stimulated Swiss 3T3 fibroblasts, PDGFRα and β colocalized in vesicular structures, which also stained with the early endosome marker EEA1 (Supplementary Fig. 3a), indicating that these receptors follow the same endocytosis pathway at least under the present experimental condition. Immunostaining of serum-starved fibroblasts revealed GLUT1 localization at the plasma membrane, observed as entire cellular shapes because of the thinness of the cells, and in perinuclear punctate structures, whereas stimulation with 50 ng/ml PDGF-BB for 10 min resulted in relocation of GLUT1 to cytoplasmic vesicular structures that merged with PDGFRα and PDGFRβ (Fig. 2a, Supplementary Fig. 3b). Colocalization was also observed when cell surface GLUT1 was pre-labeled before PDGF stimulation by capitalizing on recombinant version of the GLUT1-binding region of human T cell leukemia virus envelope glycoprotein[14] (Fig. 2b). In concert, these data show that GLUT1 is co-endocytosed with PDGFRα/β following ligand stimulation.

HK1 and HK2 catalyze the first reaction of glycolysis, phosphorylation at the position 6 hydroxyl group of glucose. Both localize to the cytoplasmic surface of the mitochondrial outer membrane colocalizing with TOMM20 (Supplementary Fig. 3c), although a fraction of HK2 also is reported to reside in the cytoplasm[15]. Co-staining of PDGFRα and HK1 or HK2 in PDGF-stimulated fibroblasts did not reveal ligand-dependent relocation of HKs (Supplementary Fig. 3c); instead, we observed that some PDGFRα-containing vesicles seemingly localized in proximity to mitochondria. Recent studies have reported inter-organelle interactions between endosomes/lysosomes and mitochondria through RAB7, MFN2, VPS13, and/or VDAC[16-19], which were also enriched in our endocytic vesicle fractionation (Supplementary Fig 1e, Supplementary Data 1). We therefore asked whether endocytic vesicles and mitochondria interact after PDGF-stimulation by employing proximity ligation assay (PLA), which detects different antibodies that reside within 40 nm of each other[20]. Hereafter, we targeted PDGFRβ for PLA because of the antibody applicability. PLA signals between PDGFRβ and HK1 or HK2, respectively, were enhanced significantly following PDGF-stimulation (Fig. 2c), indicating that PDGFR-containing endocytic vesicles are transported to within nanometers of mitochondria. Importantly, PLA signals were abolished in cells treated with *Hk1* or *Hk2* siRNAs, respectively (Supplementary Fig. 3d).

GPI and TPI1 were observed predominantly in the cytoplasm in serum-starved cells, but PDGF-stimulation resulted in their relocation to PDGFRα-containing vesicles (Fig. 2d). Other glycolytic enzymes, including ALDOA, GAPDH, PGK1, ENO1, and PKM1/2 showed strong cytoplasmic staining regardless of growth factor stimulation, but also partial co-localization to the periphery of PDGFR-containing vesicles (Supplementary Fig. 4a) in PDGF-stimulated cells. PDGF-stimulation significantly increased PLA signals between PDGFRβ and these enzymes (Fig. 2e, Supplementary Fig. 4b). Importantly, PDGF stimulation of serum-starved fibroblasts did not increase the levels of the glycolytic enzymes (Supplementary Fig. 4c). We could not evaluate the localization of PFKL, PGAM1, or PKLR owing to a lack of antibodies available for immunostaining. Collectively, however, our results show that GLUT1 is co-endocytosed with PDGFR upon PDGF stimulation and suggest that these endocytic vesicles localize close to mitochondrial hexokinases and most, if not all, other glycolytic enzymes.

### PDGF-evoked GLUT1-mediated cellular glucose uptake is dependent on receptor endocytosis

Cell surface expression of glucose transporters is thought to determine the rate of extracellular glucose uptake. Accordingly, PDGF-induced GLUT1 internalization as described might be expected to decrease cellular glucose uptake (and thereby glycolysis). Yet it has long been known that growth factor stimulation rapidly augments glucose uptake and glycolysis[5,21,22]. To explore this apparent paradox, we tested glucose uptake by Swiss 3T3 fibroblasts by incubating them

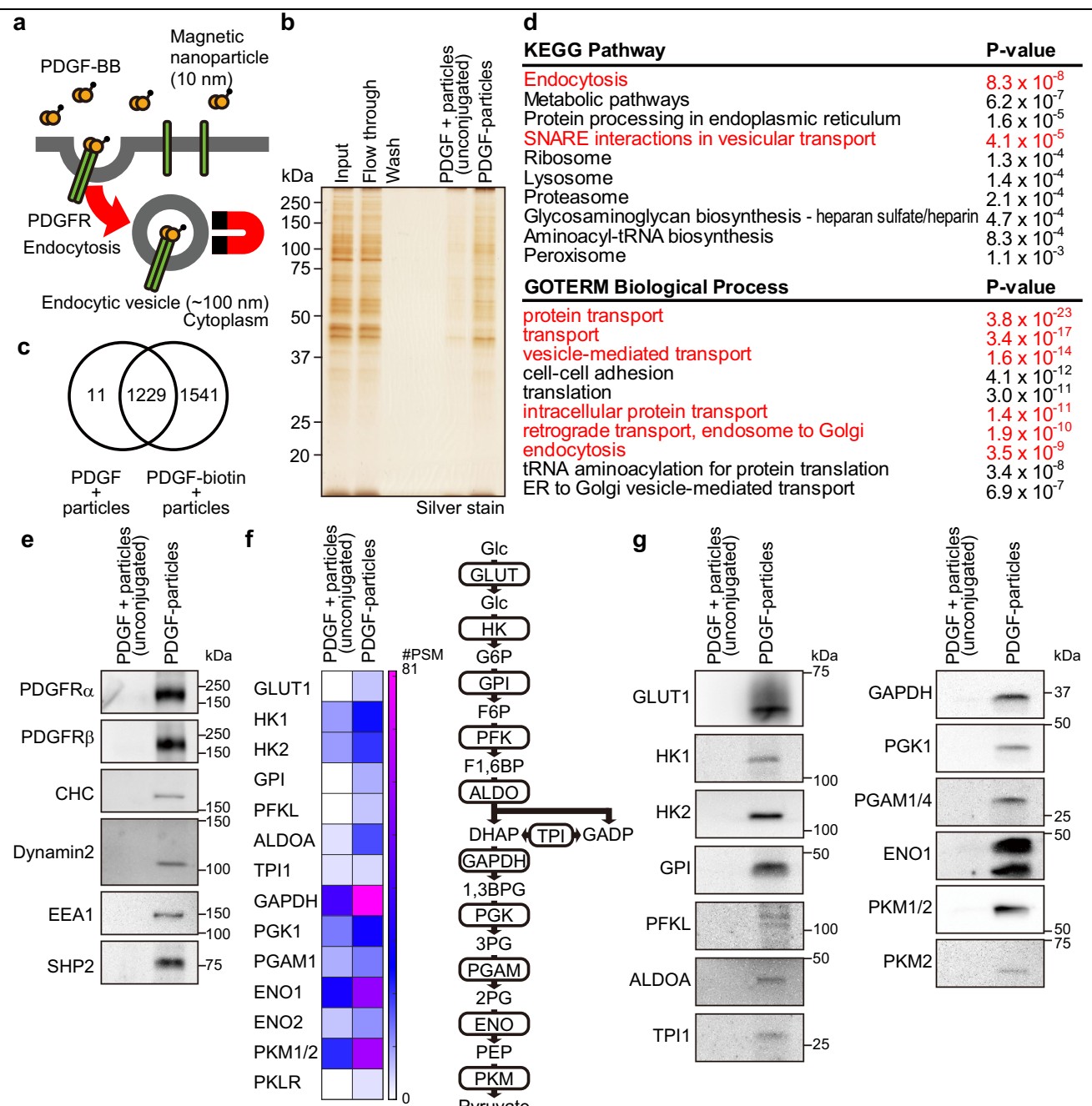

**Fig. 1 | Isolation of PDGFR endocytic vesicles utilizing magnetic nanoparticles.**
**a** Schematic showing strategy for endocytic vesicle isolation. **b** Fractions magnetically isolated from post-nuclear supernatants of PDGF-BB-biotin-conjugated nanoparticle-treated (PDGF-particle) or unconjugated PDGF-BB plus nanoparticle-treated (control) Swiss 3T3 fibroblasts were analyzed by SDS-PAGE and silver staining. Post-nuclear supernatant (input), flowthrough, and wash fractions were from PDGF-BB-biotin-conjugated nanoparticle-treated cells. A representative image from one of 5 independent experiments is shown. **c** Venn diagram shows numbers of proteins identified in the control and endocytic vesicle fractions by LC-MS/MS, as detailed in Supplementary Data 1. Data are from a single experiment. **d** Gene ontology (GO) enrichment analyses of proteins uniquely identified in the endocytic vesicle fraction. Annotations related to endocytosis or vesicle transport are in red. One-sided P values were calculated using Fisher's Exact test. **e** Fractions magnetically isolated from post-nuclear supernatants of PDGF-BB-biotin-conjugated nanoparticle-treated (PDGF-particle) or unconjugated PDGF-BB plus nanoparticle-treated (control) Swiss 3T3 fibroblasts were subjected to immunoblotting with the indicated antibodies. Representative images are shown from one of 2 independent experiments. **f** Heatmap representing numbers of peptide spectrum matches (#PSMs) of glycolysis-related proteins obtained by LC-MS/MS. A schematic of glycolysis is shown on the right. **g** Fractions magnetically isolated from post-nuclear supernatants of PDGF-BB-biotin-conjugated nanoparticle-treated (PDGF-particle) or unconjugated PDGF-BB plus nanoparticle-treated (control) Swiss 3T3 fibroblasts were subjected to immunoblotting with indicated antibodies. Representative images are shown from one of 2 independent experiments. Source data are provided as a Source Data file.

in the presence of the non-metabolizable analog 2-deoxyglucose (2DG) and measuring the accumulation of 2-deoxyglucose-6-phosphate (2DG6P). Indeed, PDGF stimulation for 10 min nearly doubled glucose uptake (Supplementary Fig. 5a). Basal and stimulation-evoked elevation of glucose uptake in Swiss 3T3 fibroblasts were dependent

on GLUT1 because treatment with 50 nM BAY876, a specific GLUT1 inhibitor at this concentration[23], almost completely abolished 2DG6P formation without affecting PDGFR tyrosyl phosphorylation (Fig. 3a, Supplementary Fig. 5b). RNA-interference experiments showed that suppression of both PDGFRα and PDGFRβ is necessary to inhibit PDGF-

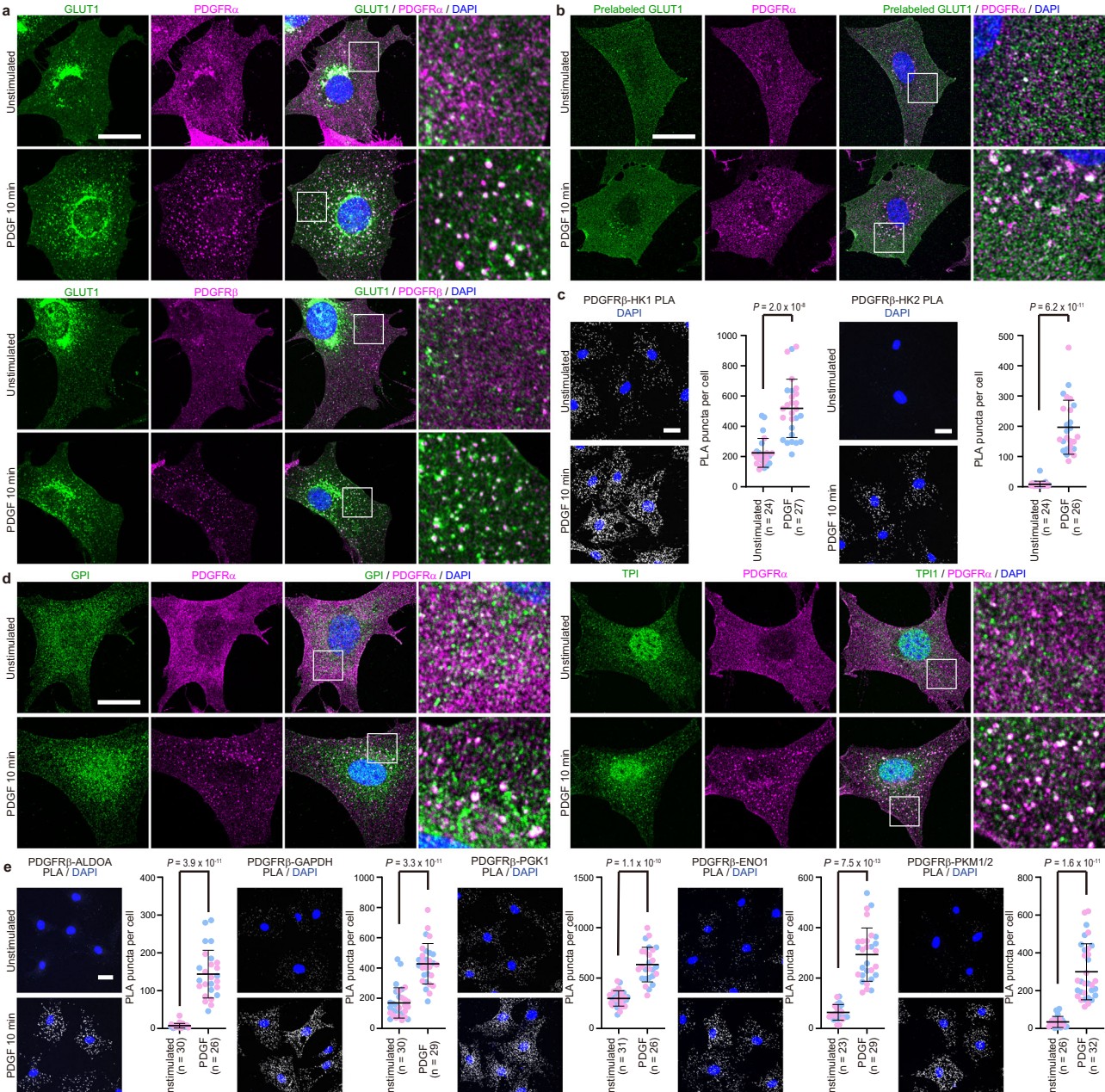

**Fig. 2 | GLUT1 and glycolytic enzymes localize to PDGFR endocytic vesicles.**
**a** Serum-starved Swiss 3T3 fibroblasts were stimulated with PDGF-BB (50 ng/ml) for 10 min or left unstimulated, and immunostained with anti-GLUT1 (green) and anti-PDGFRα or PDGFRβ (magenta) antibodies. Nuclei were stained with DAPI (blue). Higher magnification images of the boxed regions are shown. Representative images are shown for each condition from one of 2 independent experiments. **b** Cell surface GLUT1 in serum-starved Swiss 3T3 cells was labeled with recombinant GLUT1-binding region of human T cell leukemia virus envelope glycoprotein for 20 min on ice, stimulated with PDGF-BB (50 ng/ml) for 10 min at 37 °C or left unstimulated, and immunostained for surface and internalized GLUT1 (green) and PDGFRα (magenta). Nuclei were stained with DAPI (blue). Higher magnification images of the boxed regions are shown. Representative images are shown for each condition from one of 2 independent experiments. **c** Serum-starved Swiss 3T3 fibroblasts were stimulated with PDGF-BB (50 ng/ml) for 10 min or left unstimulated, and subjected to PLA with anti-PDGFRβ and anti-HK1 or -HK2 antibodies (gray). Nuclei were stained with DAPI (blue). Representative data are shown from

one of 2 independent experiments. PLA signals in the indicated number of cells were counted and plotted in the graphs. Colors of dots in the graph represent independent biological repeats. Bars represent mean ± SD of PLA signals per cell. *P* values were calculated using two-tailed Welch's *t* test. **d**. Serum-starved and PDGF-BB-stimulated or unstimulated cells were immunostained with anti-GPI or anti-TPI1 (green) and anti-PDGFRα (magenta) antibodies. Higher magnification images of the boxed regions are shown. Nuclei were stained with DAPI (blue). Representative images are shown for each condition from one of 2 independent experiments. **e** Serum-starved and PDGF-BB stimulated or unstimulated cells were subjected to PLA with anti-PDGFRβ and the indicated antibodies. Representative data are shown from one of 2 independent experiments. PLA signals were counted and plotted in the graphs. Colors of dots in the graph represent independent biological repeats. Bars represent mean ± SD of PLA signals per cell. *P* values were calculated using two-tailed Welch's *t* test. Scale bars: 20 μm. Source data are provided as a Source Data file.

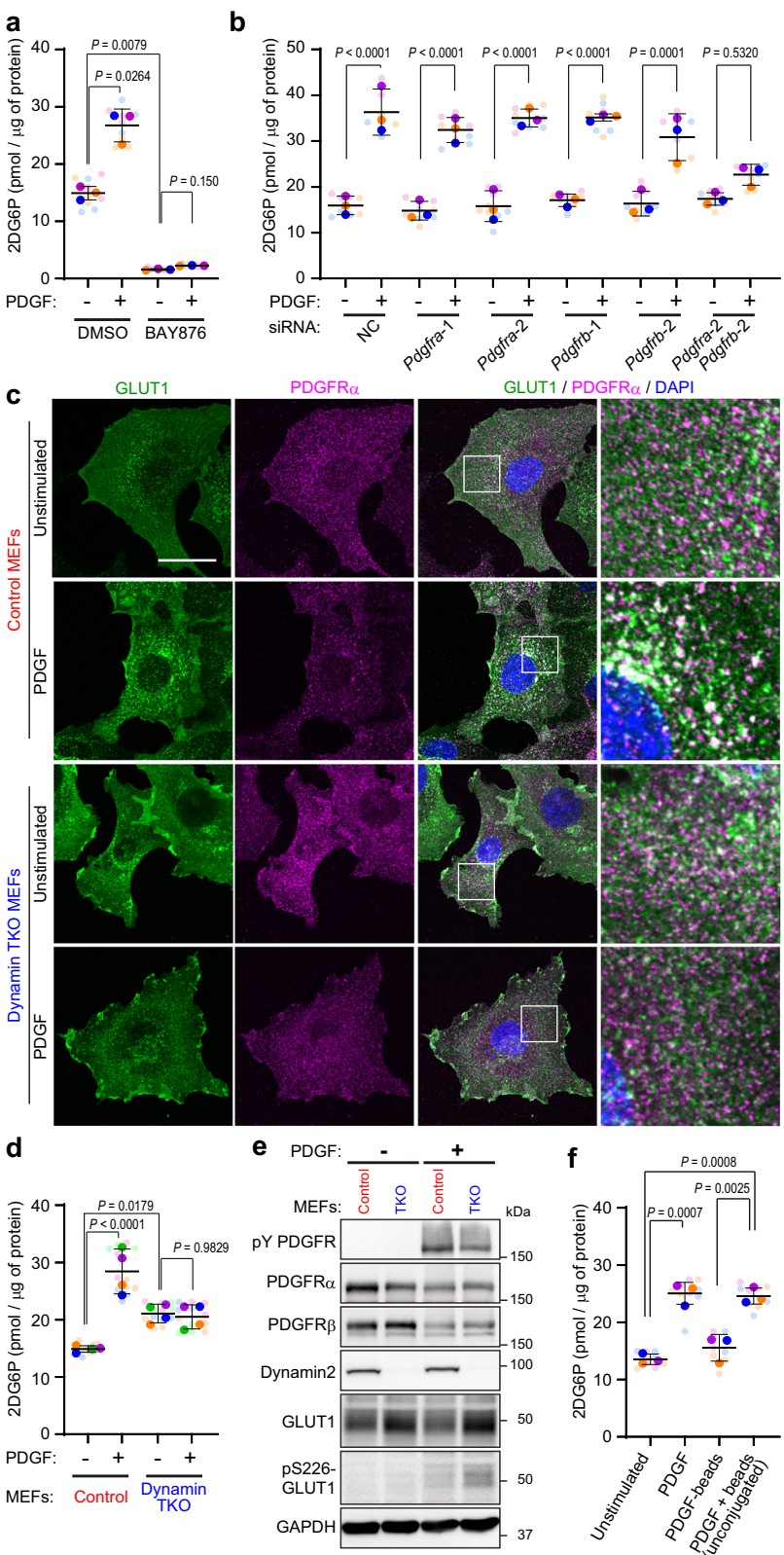

dependent glucose uptake (Fig. 3b, Supplementary Fig. 5c, d). On the other hand, surface and total GLUT1 levels decreased slightly after PDGF stimulation (Supplementary Fig. 5e, f), indicating that PDGF-dependent upregulation of glucose uptake is not due to a rapid increase in surface expression of the transporter. We conclude that PDGF elicits glucose uptake despite endocytosis of the major glucose transporter GLUT1 in 3T3 fibroblasts. Relatively unchanged surface

GLUT1 levels after PDGF stimulation in contrast to that of PDGFR, which is decreased to approximately 50% at 10 min[24], is presumably attributable to excess number of cell surface GLUT1 molecules compared to PDGFRs.

PDGFR is internalized upon ligand stimulation by clathrin-dependent and, depending on the PDGF dose, clathrin-independent endocytosis[8]. Our endocytic vesicle enrichment experiments detected

**Fig. 3 | Growth factor-evoked glucose uptake requires receptor/GLUT1 endocytosis. a** Serum-starved Swiss 3T3 fibroblasts were pre-treated with or without BAY876 (50 nM) and subjected to glucose uptake assay with PDGF-BB (50 ng/ml) in the presence of 2DG for 10 min. Graph shows 2DG6P normalized to total cellular proteins. Bars and error bars show means of biological replicates (deep-color dots, $n = 3$) and ±SD. Values of biological replicates are means of technical replicates (light-color dots). One-way ANOVA and post-hoc Tukey's test. **b** siRNA-treated Swiss 3T3 fibroblasts were serum-starved and subjected to glucose uptake assay with PDGF-BB (50 ng/ml). Graph shows 2DG6P normalized to total cellular proteins. Bars and error bars show means of biological replicates (deep-color dots, $n = 3$) and ±SD. Values of biological replicates are means of technical replicates (light-color dots). One-way ANOVA and post-hoc Tukey's test. **c** Serum-starved control or dynamin TKO MEFs were stimulated with 50 ng/ml PDGF-BB for 10 min, and immunostained with the indicated antibodies. Higher magnification images of the boxed regions are shown. Nuclei were stained with DAPI (blue). Representative data are shown from one of 2 independent experiments. Scale bar: 20 μm. **d** Serum-starved control or dynamin TKO MEFs were subjected to glucose uptake assay with PDGF-BB (50 ng/ml). Graph shows 2DG6P normalized to total cellular proteins. Bars and error bars show means of biological replicates (deep-color dots, $n = 4$) and ±SD. Values of biological replicates are means of technical replicates (light-color dots). One-way ANOVA and post-hoc Tukey's test. **e** Serum-starved control or dynamin TKO MEFs were stimulated with PDGF-BB (50 ng/ml). Lysates were subjected to immunoblotting with the indicated antibodies. Representative data are shown from one of 2 independent experiments. **f** Serum-starved Swiss 3T3 fibroblasts were subjected to glucose uptake assay with PDGF-BB (50 ng/ml), PDGF-BB-conjugated microbeads, or unconjugated PDGF and microbeads. Graph shows 2DG6P normalized to total cellular proteins. Bars and error bars show means of biological replicates (deep-color dots, $n = 3$) and ±SD. Values of biological replicates are means of technical replicates (light-color dots). One-way ANOVA and post-hoc Tukey's test. Source data are provided as a Source Data file.

clathrin heavy and light chains as well as caveolin 1 and flotillins (Supplementary Fig 1e, Supplementary Data 1), suggesting involvement of clathrin-dependent and -independent processes. Both endocytosis pathways require dynamin, the GTPase that pinches off the invaginated membrane to generate intracellular vesicles[25]. There are three dynamin isoforms, with dynamin 2 being the major isoform in fibroblasts[25,26]. To investigate the potential role of GLUT1 endocytosis in glucose uptake, we employed dynamin 1, 2, 3 conditional triple-knockout mouse embryonic fibroblasts (MEFs), which express Cre recombinase fused with estrogen receptor (ER-Cre)[26]. As expected, treatment of these cells with 4-hydroxytamoxifen (4-OHT) resulted in inducible dynamin deletion, and as expected, dynamin triple-knockout (TKO) MEFs (unlike controls) failed to show PDGF-dependent endocytosis of PDGFRα (Fig. 3c, Supplementary Fig. 6a). Notably, PDGF-stimulated GLUT1 endocytosis also failed to occur in dynamin TKO MEFs (Fig. 3c). Basal glucose uptake in TKO MEFs was enhanced compared with that in control MEFs (Fig. 3d) and was mediated by GLUT1 in both cells, as indicated by its BAY876 dependence (Supplementary Fig. 6b). By contrast, while PDGF-BB-stimulated GLUT1-dependent glucose uptake in control MEFs, PDGF failed to stimulate glucose uptake in TKO MEFs (Fig. 3d, Supplementary Fig. 6b). These findings indicate that PDGF-evoked glucose uptake requires an intact endocytosis machinery. Compared with controls, TKO MEFs had slightly reduced PDGF-BB-induced PDGFR tyrosine phosphorylation (Fig. 3e), possibly due to failure to inactivate protein-tyrosine phosphatases such as SHP2 via ligand-dependent endosomal ROS generation[24]. Nevertheless, these results exclude the possibility that the endocytosis-deficient cells are unable to respond to PDGF. The elevated basal glucose uptake in unstimulated TKO cells compared to unstimulated control cells (Fig. 3d) presumably reflects increased total and/or surface GLUT1 (Fig. 3e, Supplementary Fig. 6c), owing to a lower level of basal GLUT1 endocytosis and degradation. Additionally, the observed dynamin-dependence of PDGF-stimulated glucose uptake in MEFs was not affected by glucose concentrations in culture media (4.5 or 1 g/L) (Supplementary Fig. 6d).

Pre-treatment of Swiss 3T3 fibroblasts with the MEK inhibitor AZD6244 or the PI3K inhibitor BKM120 efficiently diminished downstream ERK1/2 or AKT phosphorylation but did not interfere with PDGF-BB-dependent PDGFR-GLUT1 co-endocytosis or impair PDGF-stimulated glucose uptake (Supplementary Fig. 7a–c). These data suggest that the RAS-ERK MAPK and the PI3K/AKT pathways, two major PDGFR-dependent signaling pathways, are not involved in regulating ligand-dependent receptor endocytosis and glucose uptake, and again indicate coincidence between GLUT1 endocytosis and enhanced glucose uptake.

The necessity of PDGFR endocytosis for stimulation-induced glucose uptake was supported further by the observation that PDGF-BB conjugated to 3 μm diameter microbeads did not stimulate glucose uptake of Swiss 3T3 fibroblasts (Fig. 3f). Beads this large cannot be incorporated into the endocytosis pathway, although they induce comparable receptor phosphorylation (Supplementary Fig. 7d). Importantly, unconjugated microbeads did not impair PDGF-BB-stimulated glucose uptake or PDGR phosphorylation (Fig. 3f, Supplementary Fig. 7d).

The finding that PDGF enhances glucose uptake in a receptor endocytosis-dependent manner, without increasing cell surface GLUT1, suggested increased activity of GLUT1 and/or HKs. A previous report indicated that PKC-stimulated phosphorylation of GLUT1 on S226 activates its transporter function and suggested an involvement of this mechanism in vascular endothelial growth factor (VEGF)-dependent glucose uptake[6]. Although PDGF-dependent GLUT1 phosphorylation at S226 was detected by using a phospho-specific antibody, phosphorylation of this site was increased in TKO MEFs, making it unlikely that S226 phosphorylation alone is responsible for PDGF-enhanced glucose uptake (Fig. 3e). To assess the alternative possibility that PDGF potentiates HK activity independent of glucose supply via transporter(s), we incubated PDGF-stimulated Swiss 3T3 fibroblasts in the presence of 1% Triton X-100, 1 mM 2DG, and 0.5 or 1 mM ATP, and quantified 2DG6P production. Under these conditions, there was no apparent effect of PDGF on HK activity (Supplementary Fig. 7e).

### Endocytic vesicles act as vehicles for glucose uptake
Taken together, our data demonstrate that PDGF enhances cellular glucose uptake via receptor endocytosis and independent of known mechanisms such as transcriptional, translational, post-transcriptional regulation of pathway component proteins, or PKC-dependent phosphorylation of GLUT1. Conceivably, PDGFR/GLUT1-positive endocytic vesicles could contribute to localized delivery of glucose to mitochondrial HKs; i.e., growth factor stimulation increases HK activity by controlling substrate access, not intrinsic enzyme activity. Endocytic vesicles are transported along microtubules by dyneins[27], and notably, cytoplasmic dynein heavy chain 1 (DYHC1) was enriched in our endocytic vesicle fractionation (Supplementary Fig. 1e, Supplementary Data 1). Treatment of Swiss 3T3 fibroblasts with the specific dynein inhibitors ciliobrevin D or dynarrestin[28] did not affect basal glucose uptake, but effectively suppressed PDGF-dependent enhancement of uptake (Fig. 4a). These dynein inhibitors suppressed PDGF-evoked PDGFRβ-HK1 PLA signals without interfering stimulation-enhanced PDGFR phosphorylation, PDGFR/GLUT1 co-endocytosis, and PDGFRβ-GAPDH proximity, also suggesting that enhanced PLA signals do not simply reflect an increase in the number of endocytic vesicles (Supplementary Fig. 8a–c). Of note, dynamin depletion or dynein inhibition suppress PDGF-dependent enhancement of glucose uptake but did not suppress basal glucose uptake (Figs. 3d and 4a), while BAY876 almost completely suppressed both basal and stimulated glucose uptake (Fig. 3a). These observations indicate the existence of an endocytosis-

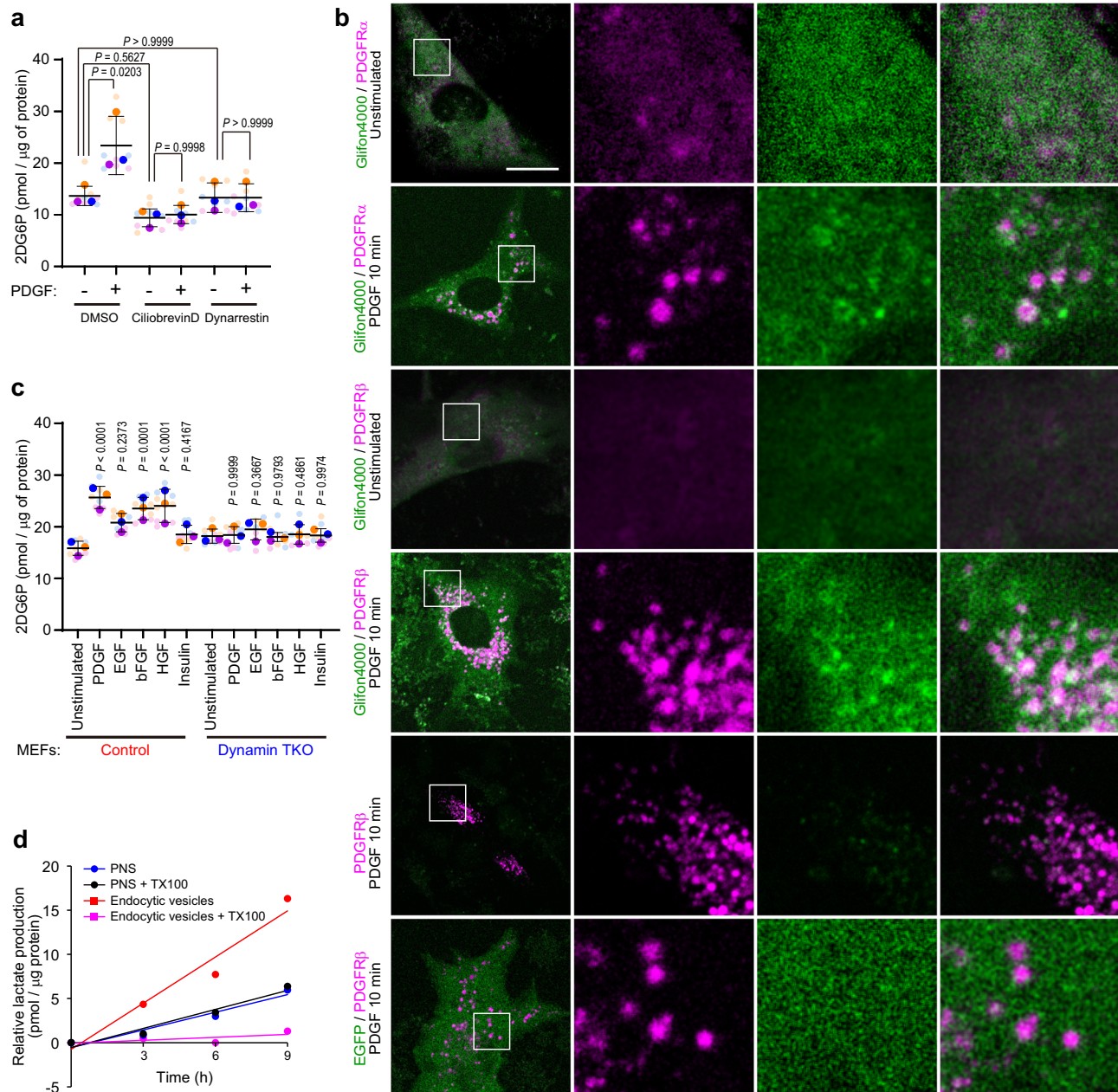

**Fig. 4 | Endocytic vesicles act as vehicles for glucose uptake. a** Serum-starved Swiss 3T3 fibroblasts were pre-treated with dynein inhibitors Ciliobrevin D (30 μM) or Dynarrestin (30 μM) for 20 min or left untreated and subjected to glucose uptake assay with or without PDGF-BB (50 ng/ml) in the presence of 2DG for 10 min. Graph shows 2DG6P normalized to total cellular proteins. Bars and error bars show means of 3 independent biological replicates (deep-color dots, biological replicates, $n = 3$) and ±SD. Values of biological replicates are means of technical replicates (light-color dots). *P* values were calculated using one-way ANOVA and post-hoc Tukey's test. **b** Swiss 3T3 fibroblasts ectopically expressing mCherry-fused PDGFR (magenta) with or without Green Glifon4000 glucose sensor or EGFP (green) were serum-starved and stimulated with PDGF-BB (50 ng/ml) for 10 min. Higher magnification images of the boxed regions are shown. Representative data

are shown from one of 5 independent live cell imaging experiments. Scale bar: 20 μm. **c** Serum-starved Swiss 3T3 fibroblasts were subjected to glucose uptake assay with PDGF-BB (50 ng/ml), EGF (50 ng/ml), bFGF (50 ng/ml), HGF (50 ng/ml), insulin (50 ng/ml) or unstimulated. Graph shows 2DG6P normalized to total cellular proteins. Bars and error bars show means of 3 independent biological replicates (deep-color dots, biological replicates, $n = 3$) and ±SD. Values of biological replicates are means of technical replicates (light-color dots). *P* values were calculated using one-way ANOVA and post-hoc Tukey's test. **d** Post-nuclear supernatants and endocytic vesicle fraction with or without 1% Triton X-100 were subjected to glycolysis assay. Graph shows lactate production normalized to protein levels in each fraction. Representative data are shown from one of 2 independent experiments. Source data are provided as a Source Data file.

and vesicle traffic-dependent, active pathway for GLUT1-mediated glucose uptake in addition to an endocytosis- and vesicle traffic-independent, passive mechanism.

To evaluate further the hypothesis that PDGFR-positive endocytic vesicles function as glucose-loaded vehicles, we employed the EGFP-derived glucose sensor Green Glifon4000, which exhibits enhanced fluorescence intensity in the presence of glucose but not of its

catabolites such as glucose-6-phosphate[29]. As GLUT1 is a facilitated glucose transporter, intra-endosomal glucose could be transported into the cytoplasm, creating local glucose gradients. Consistent with this idea, live cell imaging of PDGF-BB-stimulated Swiss 3T3 fibroblasts ectopically expressing cytoplasmic Green Glifon4000 showed enhanced fluorescence at vesicles near signals from PDGFRα or PDGFRβ fused with mCherry (Fig. 4b). In addition, we confirmed the

presence of glucose in PDGFR-containing endocytic vesicles isolated by using magnetic nanoparticles only when the entire preparation was performed in the presence of GLUT1 inhibitor BAY876 (Supplementary Fig. 8d), supporting the idea that the PDGFR vesicles contain glucose and can release glucose to the cytoplasm in a GLUT1-dependent manner.

Enhanced glucose uptake was observed in fibroblasts stimulated with other growth factors including basic fibroblast growth factor (bFGF) and hepatocyte growth factor (HGF) (Fig. 4c). We also observed a trend of increase in glucose uptake in EGF-treated MEFs as previously reported (Fig. 4c). Stimulation of glucose uptake in MEFs by bFGF and HGF also required dynamin, indicating that this mechanism is common to multiple growth factors (Fig. 4c). Notably, insulin did not significantly evoke glucose uptake in these cells (Fig. 4c), possibly reflecting low insulin receptor levels, as it also did not stimulate increased total tyrosine-phosphorylation in these cells (Supplementary Fig. 8e). The idea that growth factor-mediated glucose uptake via endosomes is a general mechanism was also supported by an observation that EGF-stimulated enhanced glucose uptake in HeLa cells was efficiently inhibited by the dynein inhibitors ciliobrevin D or dynarrestin (Supplementary Fig. 8f).

In control MEFs, PLA experiments confirmed that PDGF stimulation resulted in the juxtaposition of PDGFRβ with the glycolytic enzymes HK2, ALDOA, GAPDH, PGK1, ENO1, PKM1/2, or GLUT1, as observed in Swiss 3T3 fibroblasts. By contrast, there was no PDGF-dependent increase in PLA signal between PDGFRβ and glycolytic enzymes except GLUT1 in TKO MEFs (Supplementary Fig. 9). As these cells differ primarily in their ability to enable growth factor-induced endocytosis, we conclude that PDGF-induced adjacency of glycolytic enzymes and PDGFR requires, and occurs on, receptor-containing endocytic vesicles. In agreement with this conclusion, incubation of nanoparticle-isolated endocytic vesicle fractions in vitro in the presence of glycolytic substrates, glucose, ATP, ADP, and NAD⁺, produced lactate, the end product of glycolysis, more effectively than an unenriched post nuclear supernatant (Fig. 4d). Furthermore, pre-treatment of the endocytic vesicle fraction with 1% Triton X-100 decreased lactate production (Fig. 4d). Taken together, these data support the notion that glycolytic enzymes form a multi-enzyme complex on receptor-containing endocytic vesicles to efficiently and concertedly execute the reactions of glycolysis.

## Endocytosis machinery is necessary for the cellular carbon metabolism

To investigate the biological significance of these observations, we compared the effects of glucose-limitation on the survival of control and TKO MEFs. Both cell types survived for at least 3 days in standard high glucose DMEM (4.5 g/L glucose) in the presence of 20 ng/ml PDGF-BB without pyruvate and fetal bovine serum (FBS), and both died when glucose was absent from the media (Fig. 5a). By contrast, limiting glucose from 4.5 g/L to 0.2 g/L or 0.1 g/L in the presence of PDGF for 3 days diminished the survival of the endocytosis-deficient TKO cells when compared with control cells (Fig. 5a), signifying that dynamin depletion can induce cell death that is related to cellular glucose metabolism.

To gain further insight into the metabolic consequences of dynamin depletion, we quantified 116 cellular carbon metabolism-related substances in control and TKO MEFs cultured under glucose-limiting conditions (0.1 g/l glucose, without pyruvate) in the presence of PDGF-BB for 24 h, a time at which their viability is comparable. Metabolite levels were normalized to cell number at the time of harvest (Supplementary Data 2). Importantly, expression levels of glycolytic enzymes were comparable between control and dynamin TKO MEFs in this condition (Supplementary Fig. 10a). Moreover, total PDGFRs and total phosphorylation of PDGFRs, as well as cell surface PDGFRα were not dramatically changed between control and dynamin TKO MEFs

(Supplementary Fig. 10a, b). Furthermore, PDGFRα and GLUT1 were co-localized at vesicular structures in control but not dynamin TKO MEFs (Supplementary Fig. 10c), strongly suggesting that PDGFR-GLUT1 co-endocytosis continued to occur in control MEFs. Principal component analysis and unsupervised hierarchical clustering separated the control and TKO MEF samples ($n = 3$ each) by their genotypes (Fig. 5b, c). Twenty-seven metabolites showed statistically significant alteration between groups, while 13 were detectable only in control or TKO MEF samples (Supplementary Data 2). Notably, multiple glycolytic intermediates, including glucose-6-phosphate (G6P), fructose-6-phosphate (F6P), fructose-1,6-bisphosphate (F1,6P), diphosphoglycerate (including 1,3-bisphosphoglyceric acid; 1,3BPG), 3-phosphoglyceric acid (3PG), 2-phosphoglyceric acid (2PG), and phosphoenolpyruvic acid (PEP), were strongly diminished in TKO MEFs (Fig. 5d, Supplementary Data 2). The pentose phosphate pathway (PPP) also was affected dramatically, with 6-phosphogluconic acid (6-PG), ribose 5-phosphate (R5P), sedoheptulose 7-phosphate (S7P), and erythrose 4-phosphate (E4P) significantly decreased or showing a trend towards decrease (Supplementary Fig. 11a). Pyruvate, acetyl-CoA, and citric acid were nominally, but not statistically diminished, potentially reflecting compensatory amino acid uptake/metabolism. However, the downstream TCA cycle components, cis-aconitate, α-ketoglutarate (2-oxoglutarate; 2OG), succinate, and maleic acid were significantly decreased in TKO MEFs (Fig. 5d, Supplementary Data 2), indicating compensation is incomplete. Despite depression of these pathways, cellular ATP levels were maintained in TKO MEFs after 24 h of glucose depletion to 0.1 g/L (Fig. 5e, Supplementary Data 2). The NADH/NAD⁺-ratio was unaffected, while consistent with the observed decrease in PPP activity, NADPH/NADP⁺ was diminished in the dynamin TKO MEFs (Supplementary Fig. 11b). Other metabolomic characteristics of dynamin TKO MEFs compared to controls included increased purine metabolic intermediates, unaffected glutathione, nicotinamide, and lipid metabolism, and altered amino acid levels (Supplementary Data 2, Supplementary Fig. 11c). Overall, metabolic profiling demonstrated an important role for dynamin, and presumably, dynamin-dependent endocytic events, in central carbon metabolism pathways, including glycolysis, the pentose phosphate pathway, and the TCA cycle. Most likely, these alterations reflect a requirement for dynamin/dynamin-mediated endocytosis in receptor-stimulated glucose uptake, and are essential for survival in limiting glucose.

## Discussion

The current textbook model for glycolysis holds that extracellular glucose molecules are transported through the plasma membrane and then subjected to metabolic reactions in the cytoplasm. Regulation of this process is believed to involve quantitative or qualitative changes in cell surface transporters or glycolytic enzymes. In this study, we found that glucose uptake via GLUT1, heretofore believed to be, with few exceptions, a constitutive, basal glucose transporter, can be enhanced by growth factor action that involves cellular membrane dynamics, particularly endocytosis and vesicle trafficking. Our data suggest that glucose in the endocytic vesicles is released into the cytoplasm near the mitochondria through co-endocytosed vesicular GLUT1. Although we cannot exclude the possibility that PDGFR evokes an as yet unknown endocytosis-dependent signaling event that enhances the activity of cell surface GLUT1, we favor a model in which the endocytosis machinery plays a critical role in facilitating re-localization of GLUT1/glycolytic enzymes. We envision two potential mechanisms whereby endocytosis might promote GLUT1-dependent glucose uptake. First, endocytic vesicles could serve as platforms that activate GLUT1 and/or glycolytic enzymes. Although phosphorylation on S226 of GLUT1 per se was insufficient to induce glucose uptake in the context of PDGF stimulation, the signaling enzymes that are enriched on the endocytic vesicles (Supplementary Fig. 1f, Supplementary Data 1) could contribute to functional modification of glycolysis-related

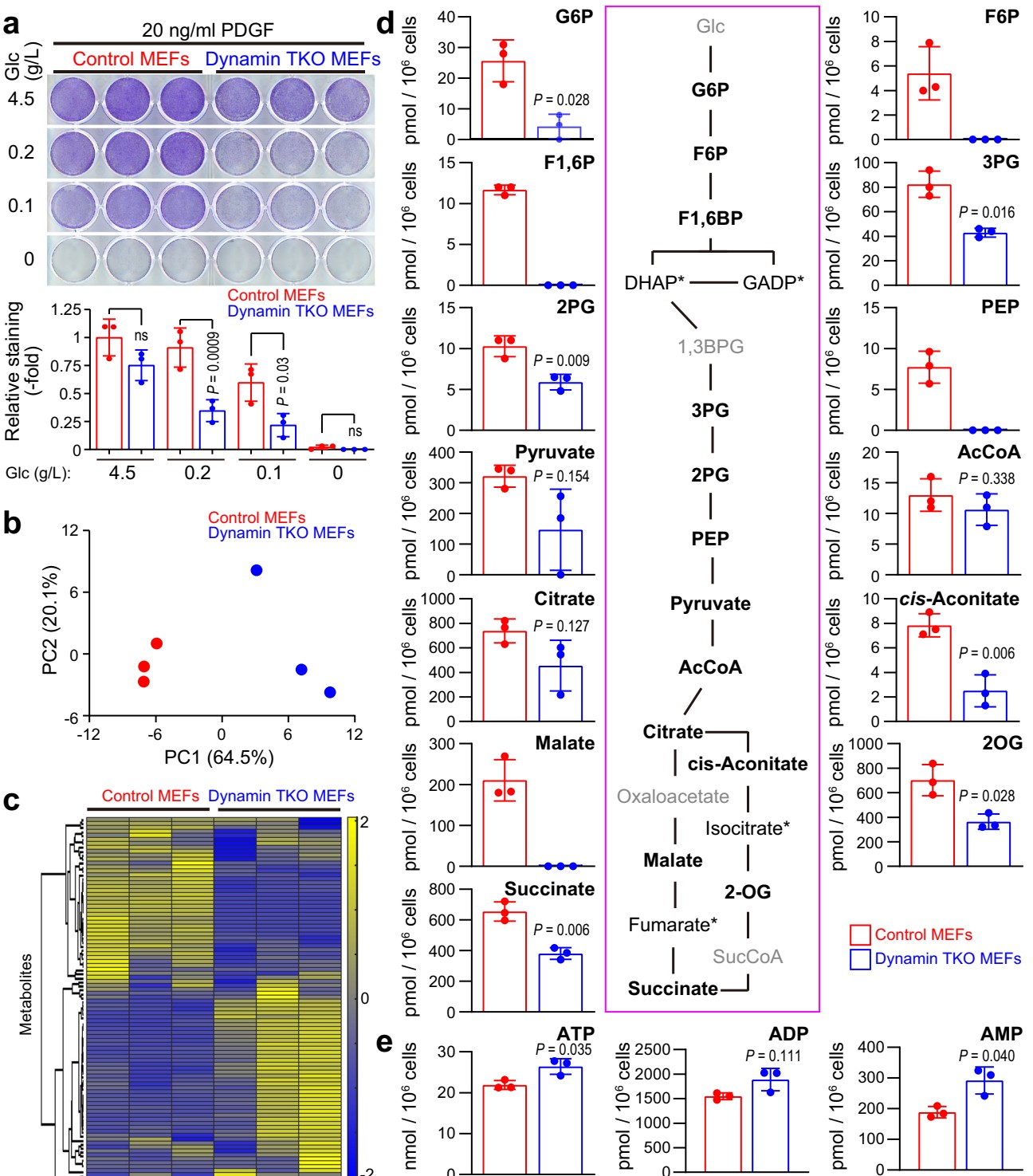

**Fig. 5 | Cellular carbon homeostasis in glucose-limited condition requires receptor endocytosis-dependent glucose uptake. a** Conditional Dynamin TKO MEFs treated with 4-OHT (Dynamin TKO) or untreated (control) were cultured in DMEM containing the indicated concentrations of glucose without FBS and pyruvate in the presence of PDGF-BB (20 ng/ml) for 3 days. Surviving cells were visualized by staining with crystal violet. Graph shows relative staining intensity in 3 wells (technical replicates, $n = 3$), with the value of control cells set as 1. Error bars represent ±SD. $P$ values were calculated by one-way ANOVA and post-hoc Tukey's test. Representative data are shown from one of 2 independent experiments. **b** Control MEFs or dynamin TKO MEFs were cultured in DMEM containing 0.1 g/ml glucose without FBS and pyruvate in the presence of PDGF-BB (20 ng/ml) for 24 h. Cellular metabolites were extracted and analyzed by mass spectrometry. Graph shows principal component analysis. **c** Heatmap shows unsupervised hierarchical clustering based on levels of 116 metabolites. **d** Graphs show average amount of each metabolite in the 3 samples (technical replicates, $n = 3$), adjusted by cell numbers. Error bars represent ±SD. $P$ values were calculated using two-tailed Welch's $t$ tests. Scheme shows glycolytic and TCA cycle metabolites. Metabolites not tested are labeled in gray, and asterisks indicate metabolites that were not detected in any sample. **e** Graphs show average amount of each metabolite in 3 samples (technical replicates, $n = 3$), normalized to cell number. Error bars represent ±SD. $P$ values were calculated using two-tailed Welch's $t$ tests. Data was obtained from a single experiment. Source data are provided as a Source Data file.

transporter and/or enzymes. Additionally, endocytosis-dependent protein-protein or protein-lipid interactions on the endosomes might promote transporter/enzyme activity. Notably, direct activation of HK1 on the mitochondria by an isoform of KRAS was reported previously[30]. Furthermore, modifications of phosphatidylinositol can occur on endocytic vesicles[31,32], and we see enrichment of phosphatidylinositol-related enzymes in our vesicle preparation (Supplementary Fig. 1f, Supplementary Data 1). Alternatively, induced proximity of glucose-loaded cargos near mitochondria and release of glucose to the adjacent cytoplasmic space might lead to accelerated G6P production simply because of the increase in local glucose concentration, given that basal intracellular glucose is quite low and diffusion in the cytosol could be too slow for efficient activation of glycolysis, perhaps because of molecular crowding[33]. The presence of a glycolytic enzyme complex, implied by this study and hypothesized previously to exist in neurons and in muscle[34,35], might enhance glycolysis by efficiently delivering substrate and concentrating intermediates locally, thereby biasing reactions that are largely at equilibrium under standard conditions. The mechanism for formation of glycolytic enzyme complex adjacent to endocytic vesicles remains to be elucidated. There may be scaffold proteins like Band 3 (SLC4A1), which targets glycolytic enzymes to the plasma membrane in erythrocytes[36], although it suppresses glycolysis and is not in our list of endocytic vesicle proteins.

Growth factors are cues for diverse cellular processes including proliferation, survival, differentiation, morphogenesis, and movement. Immediate enhancement of glucose uptake/glycolysis could prepare cells for functions that likely require a significant amount of energy. The relative importance of ligand- and endocytosis-dependent immediate upregulation and transcription/translation-dependent long-term enhancement of glycolysis may differ depending on the precise cellular context. Notably, our endocytic vesicle enrichment method detected several SLC family transporters including those for amino acids, purines, and ions (Supplementary Data 1). These findings raise the possibility that endocytic vesicles comprise a third general nutrient intake system using membrane dynamics, in addition to pinocytosis and macropinocytosis[37]. In particular, previous studies suggest decreases in glucose uptake in dynamin-deficient pancreatic β cells and T lymphocytes[38–40]. Although the primary mechanisms identified were deficient insulin secretion in β cells and decreased GLUT1 expression in T lymphocytes, receptor endocytosis-dependent glucose uptake may be involved in these processes in part. Regardless, further studies are required to clarify the involvement of receptor endocytosis in the control of cellular carbon metabolism, including its generality in vivo.

## Methods

### Antibodies, growth factors, and chemical compounds

Rabbit monoclonal anti-EEA1 (C45B10, #3288, IB 1/1000, IF 1/200), anti-Hexokinase I (C35C4, #2024, IB 1/1000, IF 1/200, PLA 1/200), anti-GAPDH (D16H11, #5174, IB 1/1000, IF 1/200, PLA 1/200), anti-PKM 1/2 (C103A3, #3190, IB 1/1000, IF 1/100, PLA 1/100), anti-PKM2 (D78A4, #4053, IB 1/1000), anti-pY849 PDGFRα/pY857PDGFRβ (C43E9, #3170, IB 1/1000), anti-AKT (C67E7, #4691, IB 1/1000), anti-pT308 AKT (D25E6, #13038, IB 1/1000), anti-ERK1/2 (137F5, #4695, IB 1/1000), and anti-phospho ERK1/2 (D13.14.4E, #4370, IB 1/1000) antibodies were purchased from Cell Signaling Technology. Rabbit monoclonal anti-GLUT1 (ab115730, IB 1/1000, IF 1/500, PLA 1/500), anti-Hexokinase II (ab209847, IB 1/1000, IF 1/100, PLA 1/100), anti-PFKL (ab181064, IB 1/1000), anti-PGK1 (ab199438, IB 1/1000, IF 1/250, PLA 1/250), and anti-ENO1/2/3 (ab189891, IB 1/1000, IF 1/500, PLA 1/500) antibodies were from Abcam. Goat polyclonal anti-PDGFRα antibodies (AF1062, IB 1/1000, IF 1/200) were purchased from R&D systems. Rabbit polyclonal anti-pS226 GLUT1 antibody (ABN991, IB 1/200) and mouse anti-phosphotyrosine monoclonal antibody cocktail 4G10 Platinum (05-

1050, IB 1/1000) were purchased from Millipore. Recombinant polyclonal anti-ALDOA antibody cocktail (711764, IB 1/200, IF 1/100, PLA 1/100) was purchased from Invitrogen. Goat polyclonal anti-PDGFRβ (sc-1627, IB 1/1000, IF 1/200, PLA 1/200), anti-EEA1 (sc-6414, IF 1/200), anti-clathrin heavy chain (sc-6579, IB 1/1000), anti-dynamin II (sc-6400, IB 1/1000), and rabbit polyclonal anti-PDGFRβ (sc-432, IF 1/200) antibodies were purchased from Santa Cruz Biotechnology but are currently discontinued. Mouse monoclonal anti-SHP2 (B-1, sc-7384, IB 1/1000), anti-TPI1 (H-11, sc-166785, IB 1/1000, IF 1/200), anti-GPI (H-10, sc-365066, IB 1/1000, IF 1/200), and anti-PGAM1/4 (D-5, sc-365677, IB 1/1000) antibodies were purchased from Santa Cruz Biotechnology. Alexa Fluor-conjugated anti-TOMM20 mouse monoclonal antibody (ab309166, IF 1/250), Alexa Fluor-conjugated donkey polyclonal anti-goat IgG (ab150130, IF 1/20000), anti-mouse IgG (ab150105, IF 1/20000), and anti-rabbit IgG (ab150073, IF 1/20000) secondary antibodies were purchased from abcam. All antibodies were used at the concentrations recommended by their manufacturers and dilutions for each experiment. EGFP-conjugated GLUT1 ligand (receptor binding domain of the human T cell leukemiavirus (HTLV) envelope glycoprotein) was purchased from Metafora Biosystems.

Biotinylated recombinant human PDGF-BB (BT220) and EGF (236-EG) were purchased from R&D Systems. Recombinant human PDGF-BB (500-P47), basic FGF (100-18B), and HGF (100-39H) were purchased from Peprotech. Recombinant human insulin (099-06473) was purchased from FUJIFILM Wako Chemicals. Streptavidin-coated magnetic iron oxide nanoparticles (SHS-10-05) and MS300/streptavidin Magnosphere microbeads (J-MS-S300S) were purchased from Ocean NanoTech and from JSR Corporation, respectively. 4-hydroxytamoxifen (4-OHT, 17308) was purchased from Cayman Chemical; BAY876 (6199) was purchased from Tocris Bioscience; Selumetinib (AZD6244, S1008) was purchased from Selleck; Buparlisib (BKM120, HY-70063) was purchased from MedChemExpress; ciliobrevin D (250401) and dynarrestin (SML2332) were purchased from Sigma-Aldrich.

### Plasmids

pcDNA3.1/Green Glifon4000 (addgene #126208) was kindly provided by Dr. Kitaguchi (Tokyo Institute of Technology). Expression vectors for mouse PDGFRα and PDGFRβ were generated from pcDNA5FRT-EF-Pdgfra-EGFPN (Addgene #66787) and pcDNA5FRT-EF-Pdgfrb-EGFPN (Addgene #66787), kindly provided by Dr. Pedersen (University of Copenhagen), by replacing EGFP-coding sequences with mCherry-coding sequences using In-Fusion® HD Cloning Kit (Clontech).

### Cell culture and transfection

Swiss 3T3 fibroblasts (3T3 Swiss albino) and Dynamin TKO ($Dnm1^{fl/fl}$, $Dnm2^{fl/fl}$, $Dnm3^{fl/fl}$,) MEFs expressing Cre-ER$^{Tam26}$, and HeLa cells were cultured in Dulbecco-modified Eagle's medium (DMEM) supplemented with 10% fetal bovine serum (FBS). To induce deletion of dynamin genes, dynamin TKO MEFs were treated with 1 μM 4-OHT for 3 days, followed by additional culture for 7 days before experimentation. Cells without 4-OHT treatment were used as controls. Swiss 3T3 cells were from JCRB Cell Bank (JCRB9019, National Institutes of Biomedical Innovation, Health and Nutrition, Japan), and Dynamin TKO MEFs were kindly provided by Dr. De Camilli (Yale University). Cells were not authenticated after receipt. HeLa cells were maintained in the Sakai Laboratory and were not authenticated. All cells were confirmed as mycoplasma-negative by the PCR method as reported previously[41].

Swiss 3T3 cells were reverse transfected with the indicated siRNAs using Lipofectamine RNAiMAX (Thermo Fisher), according to the manufacturer's protocol. Forty-eight hours post-transfection, cells were re-seeded for experiments. Pre-designed siRNAs targeting mouse $Slc2a1$ (SASI_Mm01_00068163, SASI_Mm01_00068164), $Hk1$ (SASI_Mm01_00022875, SASI_Mm01_00022876), $Hk2$ (SASI_Mm01_001

72801, SASI_Mm02_00322790), *Gpi* (SASI_Mm01_00106202, SASI_Mm01_00106203), *Aldoa* (SASI_Mm01_00058380, SASI_Mm01_000 58381), *Tpi1* (SASI_Mm01_00118126, SASI_Mm01_00118127), *Gapdh* (SASI_Mm01_00189762), *Pgk1* (SASI_Mm01_00039218, SASI_Mm01_0 0039219), *Eno1* (SASI_Mm01_00095724, SASI_Mm02_00326809), *Pkm1* (SASI_Mm01_00036290, SASI_Mm01_00036292), *Pdgfra* (SASI_ Mm02_00299581, SASI_Mm02_00299582), *Pdgfrb* (SASI_Mm01_000 97279, SASI_Mm01_00097286), and negative control siRNA (SIC001) were purchased from Sigma-Aldrich.

### Electron microscopy

Swiss 3T3 cells ($4 \times 10^5$ per dish) were seeded in a 60-mm dish, and then serum-starved for 16 h. Streptavidin-coated iron oxide nanoparticles (final concentration 2 nM) and biotin-PDGF-BB (final concentration 14 nM) were mixed in serum-free DMEM at 37 °C for 10 min. Cells were stimulated by incubating in the PDGF/nanoparticle-containing DMEM for the indicated times. Cells were then rinsed with PBS and fixed in fixative containing 2.5% glutaraldehyde and 2% paraformaldehyde at room temperature (RT), then scraped and pelleted into microtubes, and consecutively fixed overnight at 4 °C, Cells were post fixed in 1% $OsO_4$, dehydrated in a series of ethanol solutions (30%, 50%, 70%, 85%, 95%, 100%), and embedded in EMbed812 epoxy resin (Electron Microscopy Sciences, Hatfield, PA). 70 nm ultrathin sections were cut, mounted on copper grids and stained with uranyl acetate and lead citrate by standard methods. Grids were viewed using a Philips CM12 TEM (Philips) transmission electron microscope and photographed with Gatan 4k x 2.7k digital camera (Gatan Inc.).

### Isolation of PDGFR endocytic vesicles using magnetic nanoparticles

Swiss 3T3 cells ($2.5 \times 10^6$ per dish) were seeded in 150-mm dishes, and then serum-starved for 16 h. Streptavidin-coated iron oxide nanoparticles (final concentration 2 nM) and biotin-PDGF-BB (final concentration 14 nM) were mixed in serum-free DMEM at 37 °C for 10 min. Three 150-mm dishes of cells were stimulated by incubating in the PDGF/nanoparticle-containing DMEM for 5 min. Cells were then rinsed with ice-cold acidic solution (20 mM acetic acid, 150 mM NaCl) once, followed by rinsing with ice-cold PBS twice. After aspirating the PBS, cells were scraped in homogenization buffer (10 mM HEPES pH7.4, 100 mM NaCl, 1 mM EDTA, 10 mM NaF, 10 mM β-glycerophosphate, 2 mM $Na_3VO_4$) on ice, centrifuged $800 \times g$ for 3 min at 4 °C, and re-suspended in 2 ml of homogenization buffer. The plasma membrane was ruptured by nitrogen cavitation or by 30 strokes of Dounce homogenizer (tight pestle). Post-nuclear supernatant (PNS) was prepared by centrifugation of the homogenate $800 \times g$ for 5 min at 4 °C. A magnetic MS column (Miltenyi Biotec) was set on a magnet and prewashed with 500 µl of homogenization buffer. PNS was applied to the column, followed by washing the column with 3 ml of wash buffer (10 mM HEPES pH7.4, 100 mM NaCl). The column was detached from the magnet and the fraction was eluted with 300 µl of wash buffer by 5 strokes of the syringe plunger.

To quantify glucose in endocytic vesicle fraction, two 150-mm dishes of serum-starved Swiss 3T3 cells were pretreated with DMSO or 50 nM BAY876 for 20 min and then incubated with PDGF-nanoparticles or unconjugated PDGF and nanoparticles, followed by rinsing, homogenization, isolation using MS-columns, and elution as described above in the presence of DMSO or 50 nM BAY876. Aliquots (100 µl) of the elutes were immediately mixed with half the amount of 0.6 N HCl (50 µl) to inactivate hexokinase activity and then neutralized with 1 M Tris (50 µl). Glucose concentrations were then quantified by using Glucose-Glo™ Assay kit (Promega) according to the manufacturer's protocol. Absolute amounts of glucose were calculated based on signals of glucose standards, setting the signals of a buffer control as background.

### LC-MS/MS

Magnetically isolated fractions from Swiss 3T3 fibroblasts treated with PDGF-BB plus unconjugated nanoparticles (control) or PDGF-BB-conjugated nanoparticles (PDGF-particles) were prepared for mass spectrometry analysis ($n = 1$) as previously described[42]. In brief, the samples were reduced with 200 mM DTT at 57 °C for 1 h, alkylated with 500 mM iodoacetamide for 45 min at room temperature in the dark, and loaded immediately onto an SDS-PAGE gel and ran just passed the stacking region to remove any detergents and LCMS incompatible reagents. The gel plugs were excised, destained, and subjected to proteolytic digestion using 300 ng of sequencing grade trypsin (Promega) overnight with gentle agitation. The resulting peptides were extracted and desalted as previously described[42]. Aliquots of each sample were loaded onto a trap column (Acclaim® PepMap 100 precolumn, 75 µm × 2 cm, C18, 3 µm, 100 Å, Thermo Scientific) connected to an analytical column (EASY-Spray column, 50 m × 75 µm ID, PepMap RSLC C18, 2 µm, 100 Å, Thermo Scientific) using the autosampler of an Easy nLC 1000 (Thermo Scientific) with solvent A consisting of 2% acetonitrile in 0.5% acetic acid and solvent B consisting of 80% acetonitrile in 0.5% acetic acid. The peptide mixture was gradient eluted into the Orbitrap Fusion Lumos mass spectrometer (Thermo Scientific) using the following gradient: 5–35% solvent B in 60 min, 35–45% solvent B in 10 min, followed by 45–100% solvent B in 10 min. The full scan was acquired with a resolution of 240,000, an AGC target of 1e6, with a maximum ion time of 50 ms, and scan range of 400–1500 m/z. Following each full MS scan, MS/MS spectra were acquired in the ion trap using the following setting: AGC target of 3e4, maximum ion time of 18 ms, one microscan, 2 m/z isolation window and Normalized Collision Energy (NCE) of 32. All acquired MS2 spectra were searched against a UniProt human database using Sequest within Proteome Discoverer 1.4 (Thermo Fisher Scientific). The search parameters were as follows: precursor mass tolerance ±10 ppm, fragment mass tolerance ±0.2 Da, digestion parameters trypsin allowing 2 missed cleavages, fixed modification of carbamidomethyl on cysteine, variable modification of oxidation on methionine, and variable modification of deamidation on glutamine and asparagine. The identifications were first filtered using a 1% peptide and protein FDR cut off searched against a decoy database and only proteins identified by at least two unique peptides were further analyzed. Proteins annotated as common contaminant proteins (cRAP) were excluded and are listed in Supplementary Data 1. The following analyses were performed based on data in Supplementary Data 1.

### Gene annotation enrichment analysis

The list of uniquely detected proteins in the endocytic vesicle fraction was subjected to enrichment analysis to KEGG[43] and Gene Ontology knowledgebases[44] using DAVID Bioinformatics Resources Ver. 6.8 (Laboratory of Human Retrovirology and Immunoinformatics (LHRI))[45]. Enriched annotations with the ten lowest *P* values of each are listed in Fig. 1d.

### SDS-PAGE and Immunoblotting

Cells were lysed in SDS lysis buffer (50 mM Tris-HCl pH7.5, 100 mM NaCl, 1 mM EDTA, 1% SDS, 10 mM NaF, 10 mM β-glycerophosphate, 2 mM $Na_3VO_4$). Fractions from the vesicle isolation, or cell lysates were subjected to SDS-PAGE, followed by transfer to Immobilon-P PVDF membranes (Millipore). Membranes were blocked in 1% BSA/TBS containing 0.1% Tween20 for 30 min, and treated with primary antibodies in blocking buffer overnight at 4 °C, followed by treatment with HRP-conjugated secondary antibodies (Dako) for 1 h. Bands were visualized using Chemi-Lumi One Super or Chemi-Lumi One Ultra (Nacalai Tesque, 02230-30 and 11644-40) according to the manufacture's protocol, and images were obtained using an Luminograph I quantitative Cooled CCD camera detection system with ImageSaver6 Ver. 2.7.2 (ATTO) or an Amersham™ ImageQuant™ 800 with IQ800

Control Software Ver. 1.2.0 (Cytiva) as unsaturated 16-bit TIFF images. Silver staining of gels was performed using Pierce™ Silver Stain for Mass Spectrometry kit (Thermo Fisher, 24600).

## Immunofluorescence

Cells ($2 \times 10^4$) were seeded on 12 mm, poly-D-lysine-coated circular glass coverslips, and then were serum-starved for 16 h. After stimulation as indicated, cells were rinsed with PBS and fixed in 4% paraformaldehyde/PBS for 10 min at RT, permeabilized with 0.1% Triton X-100, 0.2% BSA/PBS for 10 min, and blocked with 1% BSA/PBS for 30 min, followed by sequential primary antibody and Alexa fluor-conjugated secondary antibody (and Alexa fluor-conjugated anti-TOMM20 antibody) treatments as indicated for 1 h each. For PLA, cells were incubated with primary antibodies, and then treated with donkey anti-goat (Duolink In Situ PLA Probe Anti-Goat MINUS, Sigma) and donkey anti-rabbit (Duolink In Situ PLA Probe Anti-Rabbit PLUS, SIGMA) secondary antibodies, followed by PLA procedure according to the manufacturer's protocol. Note that some experiments were performed using different batches of kit, and resulted in varying absolute numbers of PLA signals for the same targets, still maintaining the magnitudes of changes among different stimulation/condition.

Surface GLUT1 was labeled with a recombinant GLUT1-binding region of human T cell leukemiavirus envelope glycoprotein fused with EGFP. Cells were incubated in the diluted recombinant protein in DMEM (20-fold). To quantify cell surface GLUT1, cells were incubated in the diluted GLUT1-binding protein at 4 °C for 30 min and fixed. To observe colocalization of GLUT1 with PDGFR, serum-starved cells were first labeled with the diluted GLUT1-binding protein at 4 °C for 30 min, followed by incubation with 50 ng/ml PDGF-BB for 10 min at 37 °C, and fixation. Since the recombinant protein showed fluorescence intensity below detectable levels with our equipment on receipt, we visualized the ligand by treating it with anti-GFP antibody (MBL, #598, 1/1000) and Alexa fluor-conjugated secondary antibody. Coverslips were then mounted on glass slides using Prolongold containing DAPI (Thermo Fisher). Images were obtained using an FV1000 confocal microscopy system with ASW Ver. 1.6 (Olympus), an LSM710 confocal microscopy system with Zen 2.3 Ver. 14.0.23.201 (Zeiss), or an LSM810 confocal microscopy system with Zen (blue edition) 3.6 (Zeiss) as unsaturated 16-bit TIFF images. The confocal microscopes were set to obtain 1 Airy Unit (AU), and single-plane images are presented in figures.

## Image analysis

To assess colocalization, regions of interest (ROIs) were set manually according to cell shapes, and Pearson's correlation coefficients were calculated using the "coloc2" function of FIJI/ImageJ Ver. 1.53[46] from the indicated numbers of images per condition.

For PLA images, ROIs were set according to cell shapes and the numbers of punctate signals generated by PLA were counted using the "Find Maxima" function of FIJI/ImageJ software Ver. 1.53 from 6 to 10 images per condition.

For surface GLUT1 quantification, total fluorescence intensities, background signal intensities, and cell areas in images were obtained using FIJI/ImageJ Ver. 1.53 software from 6 to 10 images per condition. Relative fluorescence intensities per cell area were calculated by setting the average intensity of images of control cells to 1.

## Live cell imaging

Swiss 3T3 fibroblasts ($5 \times 10^4$/10 µl suspension) were transfected with a total 0.5 µg of indicated expression vectors by using a Microporator Mini (Digital Bio Technology, MP-100), setting pulse voltage 1350 V, pulse width 30 ms, and pulse number 1, according to the manufacturer's protocol. One hundred thousand ($1 \times 10^5$) of electroporated cells were then seeded onto a well of a poly-D-lysine-coated µ-Slide 8 Well high Glass Bottom chambered coverslip (ibidi GmbH, 80807). Eight hours after seeding, cells were serum-starved for following 16 h.

Cells were observed using an LSM710 confocal microscopy system with Zen 2.3 Ver. 14.0.23.201 (Zeiss) with a stage-top incubator at 37 °C in phenol red-free, HEPES-buffered DMEM (Nacalai, 09891-25). Images were obtained as unsaturated 16-bit TIFF images. The confocal microscope was set to obtain approximately 2.4 Airy Unit (AU), and single-plane images are presented in figures.

## Glucose uptake assays

Glucose uptake was quantified using Glucose Uptake-Glo™ Assay kit (Promega) according to the manufacturer's protocol with some modifications. Cells ($2 \times 10^4$ per well in 24-well plate or $1 \times 10^4$ per well in 48-well plate) were seeded and serum-starved for 16 h. Cells were then rinsed once with DMEM without FBS and glucose, and once with PBS. Cells were then incubated in PBS containing 1 mM 2DG with or without 50 ng/ml of growth factor at 37 °C for 10 min. Cellular glucose uptake was terminated by adding acidic Stop Buffer, followed by addition of Neutralization Buffer. The lysates were mixed with the 2DG6P detection reagent, containing G6P dehydrogenase, NAD$^+$, reductase, proluciferin, ATP and recombinant luciferase, and incubated at RT for 1 h. Luminescence signals were quantified using a SpectraMaxiD5 or a SpectraMaxL plate reader with Soft Max Pro 7 Ver. 7.1 (Molecular Devices). Absolute amounts of cellular 2DG6P formation were calculated based on signals of 2DG6P standards, setting the signals of 2DG-untreated cells as the background. 2DG6P formation was normalized by amounts of protein per well. For inhibitor experiments, cells were pre-treated with vehicle or indicated concentrations of BAY876, AZD6244, BKM120, ciliobrevin D, or dynarrestin for 20 min, and then subjected to rinsing and incubation with 2DG in the presence or absence of growth factor. Vehicle or inhibitor was also added to the solutions for rinsing and 2DG/growth factor treatment. For the microbeads experiment, biotinylated PDGF-BB (1.25 µg per well for 24-well plate) was mixed with MS300/streptavidin Magnosphere microbeads (2.0 µg per well for 24-well plate) in 400 µl of PBS and was incubated at RT for 30 min. Beads were then washed 5 times with 1 ml PBS and were resuspended in PBS containing 1 mM 2DG. Cells were treated with the suspension to quantify 2DG6P production.

## In vitro hexokinase assay

Cellular hexokinase activity was measured utilizing the Glucose Uptake-Glo™ Assay kit (Promega) with a modification from the protocol described above. Cells were treated with or without PDGF for 10 min, and rinsed with glucose-free DMEM and PBS. Cells were then incubated in reaction buffer (50 mM Tris–HCl pH 7.4, 10 mM MgCl$_2$, 1% Triton X-100, 1 mM 2DG) containing the indicated concentration of ATP, instead of 2DG-containing PBS, at 37 °C for 10 min. The reactions were terminated by adding Stop Buffer from the kit, and then the mixtures were neutralized. The mixtures were then mixed with the 2DG6P detection reagent and incubated at RT for 1 h. Luminescence signals were quantified using SpectraMaxiD5 plate reader with Soft Max Pro 7 Ver. 7.1 (Molecular Devices). Absolute amounts of 2DG6P production were calculated based on signals of 2DG6P standards. 2DG6P production was normalized by amounts of protein per well.

## In vitro glycolysis assay

PNS and endocytic vesicle fraction were incubated in the presence or absence of 1% Triton X-100 on ice for 10 min. Protein concentrations of PNS and endocytic vesicle fraction were quantified using DC Protein Assay kit (BIO-RAD) and CBQCA Protein Quantitation Kit (Invitrogen), respectively. Glycolysis reaction was initiated by adding 1/20 volume of PNS or the endocytic vesicle fraction to the reaction mixture (final concentration, 100 mM phosphate buffer pH 7.4, 5 mM NaCl, 2.5 mM MgCl$_2$, 0.1 mM DTT, 15 mM ADP, 1 mM ATP, 7 mM NAD$^+$, 50 mM glucose). After the indicated time periods of incubation at 37 °C, the amounts of lactate in aliquots were quantified immediately using Amplite Fluorometric L-Lactate Assay Kit (AAT Bioquest).

Fluorescence signals were quantified using a SpectraMaxiD5 plate reader with Soft Max Pro 7 Ver. 7.1 (Molecular Devices). Reaction mixtures without PNS or endocytic fraction were set as backgrounds. Absolute amounts of lactate were calculated based on signals of standards, and relative lactate productions were calculated by normalizing the values with amounts of protein.

### Cell survival assay

Control or dynamin TKO fibroblasts ($1 \times 10^5$ per well) suspended in DMEM supplemented with 10% FBS were seeded on 12-well plate and incubated at 37 °C in $CO_2$ incubator for 16 h. Media were then changed to DMEM without FBS and pyruvate containing PDGF-BB (20 ng/ml) and indicated concentrations of glucose (day 0). Cells were incubated for 3 days, and media were changed every day. On day 3, cells were rinsed with PBS, followed by staining with 0.5% crystal violet solution containing 25% methanol for 30 min. Culture plates were rinsed with tap water and air dried. Plates were then scanned as 16-bit TIFF images, and staining intensities of each well were quantified using ImageJ/FIJI software Ver. 1.53.

### Metabolomic analysis

Control or dynamin TKO fibroblasts ($1.5 \times 10^6$), suspended in DMEM supplemented with 10% FBS, were seeded on four 100-mm dishes and incubated at 37 °C in a $CO_2$ incubator for 16 h. Media were then changed to DMEM without FBS and pyruvate containing 0.1 g/L glucose. Three of the four dishes were used for metabolite extraction (technical replicates, $n = 3$), and the other was used for cell counting. Twenty-four hours after medium change, cells were washed twice with 5% mannitol solution, and then treated with 800 µl of methanol and left for 30 s. The cell extracts were then mixed with 550 µl of Milli-Q water containing internal standards (H3304-1002, Human Metabolome Technologies (HMT), Tsuruoka, Yamagata, Japan) and left for another 30 s. The extracts were centrifuged at $2300 \times g$ and 4 °C for 5 min and 800 µl of the aqueous layer was centrifugally filtered through a Millipore 5-kDa cutoff filter (UltrafreeMC-PLHCC, HMT) to remove macromolecules ($9100 \times g$, 4 °C, 120 min). The filtrate was centrifugally concentrated and re-suspended in 50 µl of Milli-Q water.

The metabolome analysis was conducted with a C-SCOPE package of HMT using capillary electrophoresis time-of-flight mass spectrometry (CE-TOFMS) for cation analysis and CE-tandem mass spectrometry (CE-MS/MS) for anion analysis. Briefly, CE-TOFMS analysis was carried out using an Agilent CE capillary electrophoresis system equipped with an Agilent 6210 time-of-flight mass spectrometer (Agilent Technologies, Waldbronn, Germany). The systems were controlled by Agilent G2201AA ChemStation software version B.03.01 for CE (Agilent Technologies) and connected by a fused silica capillary (50 µm i.d. ×80 cm total length) with commercial electrophoresis buffer (H3301-1001 and I3302-1023 for cation and anion analyses, respectively, HMT) as the electrolyte. The spectrometer was scanned from m/z 50–1000[47]. Peaks were extracted using MasterHands, automatic integration software (Keio University, Tsuruoka, Yamagata, Japan)[48] and MassHunter Quantitative Analysis B.04.00 (Agilent Technologies) in order to obtain peak information including m/z, peak area, and migration time (MT). Signal peaks were annotated according to the HMT metabolite database based on their m/z values with the MTs. The peak area of each metabolite was first normalized with respect to the area of the internal standard and metabolite concentration was evaluated by standard curves with three-point calibrations using each standard compound. Hierarchical cluster analysis (HCA) and principal component analysis (PCA) were performed by HMT's proprietary software, PeakStat and SampleStat, respectively.

### Statistics and reproducibility

No statistical method was used to predetermine sample sizes. Samples were not randomized. The investigators were not blinded to allocation during experiments or outcome assessment. Sample sizes and statistical tests for each experiment are denoted in the figure or legends. Each experiment was performed at least twice per condition of the experiment, and representative images from one of the biological replicates are shown in each panel.

Statistical analysis was performed by using the two-tailed Welch's t test (Figs. 2c, e, and 5d, e, Supplementary Fig. 11a–c, Supplementary Data 2), unpaired two-tailed t test (Supplementary Figs. 3b and 5a, e), one-way ANOVA with post-hoc Tukey's test (Figs. 3a, b, d, f, 4a, c and 5a, Supplementary Figs. 6a–c, d, 7b, e, 8d, f and 10b), Brown-Forsythe and Welch one-way ANOVA test with post-hoc Games-Howell test (Supplementary Figs. 3d, 4b, 8c and 9) by using Graphpad Prism 8 Ver. 8.4.3 software, where appropriate. One-sided Fisher's Exact test was used for gene annotation enrichment analysis in Fig. 1d. Precise P values can be found in the figures.

### Reporting summary

Further information on research design is available in the Nature Portfolio Reporting Summary linked to this article.

## Data availability

The mass spectrometric raw files for proteomic analysis are accessible at https://massive.ucsd.edu under accession MassIVE MSV000092208 and at www.proteomexchange.org under accession PXD043112. The mass spectrometric raw files for metabolomic analysis are not available due to the inclusion of the service provider's confidential analytical parameters, but the absolute amounts of metabolites calculated are shown in Supplementary Data 2. Other source data are provided with this paper as a Source Data file. Source data are provided with this paper.

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

## Acknowledgements

We thank Dr. P. De Camilli (Yale School of Medicine) for dynamin TKO MEFs. We thank Drs. T. Kitaguchi (Tokyo Institute of Technology) and L. B. Pedersen (University of Copenhagen) for plasmids. We also thank Drs. T. Akaike (Tohoku University), R. L. Possemato (NYU Grossman School of Medicine), and M.R. Philips (NYU Grossman School of Medicine) for helpful comments and discussion. This work was supported by Japan Society for the Promotion of Science (JSPS) 18H06120, 20K06633 (to R.T.), 20H00488 (to Y.S.), and, while R.T. was in the Neel laboratory, by NIH R01 CA49152 (to B.G.N.). R.T. was also supported by the Takeda Science Foundation and the Kato Memorial Bioscience Foundation. We thank NYU Langone Health DART Microscopy Lab for their assistance with TEM work. Microscopy and Proteomics labs are partially funded by NYU Cancer Center Support Grant NIH/NCI P30CA016087. The mass spectrometric experiments were supported with a shared instrumentation grant from the NIH, 1S10OD010582-01A1 for the purchase of an Orbitrap Fusion Lumos.

## Author contributions

R.T. conceptualized and designed research. R.T., B.U., F.X.L. performed research and analyzed data. R.T., B.U., F.X.L., B.G.N., R.S., Y.S. wrote and edited the paper. B.G.N., R.S., Y.S supervised research.

## Competing interests

B.G.N. is co-founder and received equity and consulting fees from Northern Biologics, Navire Pharma, Lighthorse Therapeutics and Aethon Therapeutics. He also is on the Scientific Advisory Boards of Arvinas and Recursion and received consulting fees and equity from each. His spouse own shares in Moderna. The remaining authors declare no competing interests.
