## [Peer Review File · Nature Communications]

Endocytic vesicles act as vehicles for glucose uptake in response to growth factor stimulationREVIEWER COMMENTS

Reviewer #1 (Remarks to the Author):

This manuscript explores glucose uptake in response to growth factor-stimulated endocytosis of the glucose transporter 1. The results present a novel mechanism in the regulation of glycolysis and will be of interest to a wide range of investigators studying glycolysis and/or receptor endocytosis. The experiments use appropriate controls and the data is clearly presented. The manuscript would benefit from additional rationale for experimental design, further quantification of existing data and additional experimental replicates (three independent experiments per assay). Further, the authors should test whether knockdown of PDGFRalpha and/or PDGFRbeta attenuates GLUT1 internalization and glucose uptake in response to PDGF-BB ligand treatment.

Major comments:

1. The authors do not state which PDGFR dimer they hypothesize is being activated in response to PDGF-BB ligand stimulation in their system. PDGF-BB stimulation of fibroblasts has previously been shown to result in formation of PDGFRalpha/beta heterodimers and PDGFRbeta homodimers. The authors frequently switch between assaying the two receptors - for example, the immunofluorescence assays in Figure 2 use antibodies against PDGFRalpha, while the PLA studies in the same figure use antibodies against PDGFRbeta. The authors need to clearly present their rationale for assaying one receptor over another in each experiment.
2. It is unclear from the images in Figure 3b if dynamin depletion inhibits PDGF-BB-dependent endocytosis of PDGFRalpha, as the magenta signal for the receptors has a similar expression pattern in control and TKO MEFs. The authors need to show single channel images for these experiments and quantify the extent of PDGFRalpha internalization in each case.
3. The authors focus on the co-endocytosis of GLUT1 and the PDGFRs, and a subsequent increase in glucose uptake, in response to PDGF ligand stimulation. Though Figure 3h indicates that this phenomenon may also take place in response to a subset of additional growth factors, it will be critical to the authors' hypothesis that they demonstrate that knockdown of PDGFRalpha and/or PDGFRbeta attenuates GLUT1 internalization and glucose uptake in response to PDGF-BB ligand treatment.
4. The figure legends often indicate that two independent experiments were performed, when 3 independent experiments would be more appropriate and strengthen the authors' conclusions. Further, in several instances legends indicate 3 technical replicates and 2 independent experiments. As there are only 3 datapoints on several of the relevant graphs, it is unclear what data was averaged for visualization. In instances where individual cells are analyzed (i.e. PLA), the authors are advised to use superplots to indicate individual datapoints as well as averages from independent experiments. Statistics should then be performed using the average values per experiment.

Minor comments:

1. Given the changing meaning of the label “control” throughout the manuscript, the label “PDGF-BB 0 min” might be more clear when comparing unstimulated versus ligand-stimulated cells.
2. The authors should comment on the lack of plasma membrane PDGFRalpha signal in the absence of ligand stimulation – for example, in Figures 2a,b.
3. The authors should speculate why introduction of the control siRNA in extended data Figure 3c lowers PLA puncta per cell numbers compared to Figure 2c.
4. The authors should add single channel images for the experiments in Figure 2d.
5. The conclusion “Basal glucose uptake in TKO MEFs was slightly enhanced compared with that in control MEFs (Fig. 3c)...” is misleading, as there is a statistically significant difference.
6. On page 7, line 10, change “update” to “uptake”.
7. On page 7, line 24, change “3g” to “3h”.
8. Antibody dilutions should be provided for all assays.

Reviewer #2 (Remarks to the Author):

Tsuzumi et al presents experiments demonstrating platelet derived growth factor (PDGF) induced stimulation of cells harboring PDGFR result in endocytosis of glucose transporter 1 (GLUT1) via the same endocytic vesicles. The author also present experiments suggesting evidence that critical glycolytic enzymes localize in regions of mitochondria in proximity to the endocytosed vesicles containing PDGFR/GLUT1. On the basis of the experiments presented in the manuscript the authors propose that “that PDGF stimulated glucose uptake depends on receptor/transporter endocytosis” providing evidence for “a new layer of regulation for glycolytic control governed by cellular membrane dynamics”.

Major comments:

1) The authors provide evidence for transport of GLUT1-containing vesicles to mitochondria by comparing the localization of endocytosed PDGFR, which colocalizes with endocytosed GLUT1 (Fig. 2a, b), and the hexokinases HK1 and HK2, which are known to localize to the cytoplasmic surface of mitochondria (Extended Data Fig. 2c). To complement these experiments, the authors perform a proximity ligation assay (PLA) using antibodies against PDGFRb and HK1 and HK2. The authors show that PLA signals, which indicate close proximity of PDGFR-containing vesicles to mitochondria, are significantly increased after PDGF stimulation (Fig. 2c). However, this is not surprising since the number of endocytic vesicles containing PDGFR increases with PDGF stimulation and random encounters of these vesicles with HK1 and HK2 could produce a PLA signal. As a control experiment, the authors perform PLA

after siRNA mediated HK1 and HK2 knockdown, but this merely validates that the assay works properly. How meaningful is the increase in PLA puncta per cell after PDGF stimulation? For example, does trafficking of unrelated endocytic vesicles (i.e., GLUT1-negative vesicles) also produce a PLA signal? It is important to show in some way that the PLA signal does not simply reflect an increase in the number of endocytic vesicles after PDGF stimulation.

2) The authors show that limiting glucose in the presence of PDGF (without pyruvate and FBS) induces cell death over a 3-day period more so for dynamin TKO MEFs than for control MEFs (Fig. 4a). However, it is not clear how this relates to the regulation of cellular carbon metabolism by PDGF through an endocytosis-dependent mechanism (versus a long-term transcription/translation-dependent mechanism) when PDGFR/GLUT1 trafficking is likely over after 1-hour PDGF stimulation because, as the authors clearly show, PDGFR is largely degraded by this time point (Extended Data Fig. 4d). How do the authors explain this?

Minor comment:

It is not clear in either figure legends or Methods how confocal microscopy images are presented. Please indicate whether images are single confocal planes or maximum intensity projections.

All in all, a revised manuscript following a major revision could be considered for publication

Reviewer #3 (Remarks to the Author):

The authors describe an exciting novel mechanism that allows fibroblast cell lines to boost their glucose uptake and utilization upon growth factor stimulation. Using nanoparticle-based isolation of PDGFR-positive vesicles, the authors highlight an inducible mechanism for GLUT1-endocytosis that delivers the receptor directly to glycolytic enzymes. Although the manuscript provides compelling PLA data confirming co-localization of PDGFR-containing vesicles with the glycolytic machinery, some major points need to be addressed to strengthen the hypothesis.

Major comments:

1) The author use PDGF-mediated induction of GLUT1 endocytosis to derive all their observations. It is unclear why they used this growth factor in MEFs? Is this regulation of GLUT1-endocytosis a common feature for growth factors in fibroblast cells? What do other growth factors would do, like EGF? The old paper on EGF induced glucose uptake was never followed up. Would this mechanism be also true for other cell types and other growth factors? These important questions need to be addressed.

2) The authors use high amounts of glucose (4.5 g/L) in all of their studies. In a physiological setting that would refer to a highly diabetic condition. Also, in conditions of tumor growth this concentration is way too high to be physiological. I acknowledges that these are the commonly used cell culture conditions for

such experiments, however, I question how physiological the responses really are? Does this hold true also for lower glucose concentrations, which would reflect more 5 mM of an in vivo situation?

3) How are the glycolytic proteins associated with the endocytic vesicles. Via tethers? Anything from the interactome?

4) Line 12 on page 4: The authors state that localization of HK1/2 to the mitochondrial outer membrane is shown in Ext. Fig. 2c, 3a, and 3b, although the referenced data does not show this. This needs to be addressed. Co-stainings and quantifications with mitochondrial markers, like Tom20 need to be conducted.

5) Line 15 on page 4: Where did the authors observe proximity of PDGFR α -containing vesicles to mitochondria? This data is not included in the manuscript and needs to be presented.

6) The authors should discuss their data in a broader fashion. There are studies related to dynamin 2 KO in b-cells and other metabolically active organs, which show connections to glucose uptake and insulin secretion, thus supporting this proposed connection. In addition, there are plenty of studies on endosomal regulators in mice, which reveal an impact in glucose uptake, glycolysis and glucose metabolism in general which should be incorporated into the bigger picture of the observation presented here and needs to be thought about and discussed.

7) The authors propose increased GLUT1-mediated glucose transport to cause increased intracellular glucose accumulation by ruling out enhanced glucose-retention by testing hexokinase activity. However, as GLUT1 mediates glucose transport via facilitated diffusion, the authors should assess glucose levels within their isolated vesicles. This is especially in light of increased glucose concentrations nearby vesicles, as assessed by Green Glifon4000 (Fig. 3g).

8) How to explain unchanged surface levels of GLUT1 in MEFs, given the increased endocytosis of GLUT1 together with PDGFR as shown upon PDGF treatment? In 3T3s, there is a decrease (Ext. Data Fig. 4c). Any effect on recycling of GLUT1?

9) Please adjust the order of figures with their respective labeling according to the order in which the data is referenced in the results section to allow an easier flow of reading.

Minor comments:

1) Line 24 on page 4: TPI should read TPI1.

2) Line 4 on page 6: Please correct "Fig. 3d" to "Fig. 3d".

3) Line 26 on page 6: "making it is unlikely" should read "making it unlikely".

4) Lines 11-14 on page 7: This sentence is confusing and should please be revisited.

5) Lines 19-21 on page 7: Please also show Green Glifon4000 co-staining with PDGFR α/β in cells not treated with PDGF.

6) Line 22 on page 7: "(bFGF)- and" should read "(bFGF) and".

7) Line 24 on page 7: Referenced Fig. 3g should instead reference to Fig. 3h.

8) Lines 2-4 on page 9: This conclusion is questionable, as in addition to increased AMP levels, ADP is also elevated.

Reviewer #4 (Remarks to the Author):

In their study "Endocytic Vesicles as Conduits for Glucose Uptake Triggered by Growth Factor Stimulation," Tsutsumi et al. describe a novel pathway for glucose uptake in response to growth factor signaling. Employing proteomics of endocytic vesicles, different imaging techniques, metabolite quantification, and metabolic assays, the authors state that upon activation by growth factors, cells initiate the formation of endocytic vesicles enriched with the glucose transporter Glut1, key glycolysis enzymes, and glucose itself. These specialized vesicles appear to be selectively transported into close proximity with the mitochondria. They provide data that, upon perturbation of the endocytic machinery cellular carbon metabolism is impaired.

While the finding is novel and deepens our understanding of cellular metabolism and the regulation of glycolysis, with profound implications for physiological processes, the study does bear some severe technical concerns regarding the proteomics data and certain experimental aspects. The proteomics analyses indicate that the enzymes might be enriched but it cannot be excluded that the detection is due to contaminations. Therefore, alternative approaches such as immunogold labeling electron microscopy would be required to give an ultimate prove of the concept. Furthermore, to reinforce the general applicability of this discovery, the study could benefit from the use of multiple cell lines for the main experiments or the use of alternative stimuli in addition to PDGF stimulation. This would bolster the robustness of their findings.

While the discovery of an endocytosis mediated glucose uptake is in general interesting, some conceptual questions remain. How should the HK1 and HK2 residing at the outer mitochondrial membrane access the glucose transported in the vesicles? How should the glycolytic machinery be incorporated specifically into the vesicles? For this some interaction with receptors or a sorting mechanism would be required.

Main points:

- While in the proteomics section, endocytic vesicles seem to be enriched, the high number of identified proteins indicates strong contaminations, what makes the results difficult to interpret. Moreover, the authors state that glycolytic proteins are enriched in the purified endocytic vesicles, but no enrichments towards other fractions of the purification process are shown, only peptide counts of the vesicle fraction. No general quality assessment of the proteomics data is performed. Only peptide counts for selected proteins are picked and shown as changed without any statistical testing or comprehensive data analysis. The following experiments would strongly increase the robustness of the proteomics experiments:

- o Show the enrichment of glycolytic enzymes and GLUT1 in the endocytic fraction versus the input and the flow through. The enrichment profiles of enzymes for, for example fructose metabolism, purine metabolism, other transporters or signaling complexes such as MTOR should be shown as negative controls.
 - o Purity could be increased by a density gradient or differential centrifugation before the magnetic bead-based enrichment, alternatively a second alternative purification method based on different separation techniques prior to proteomics analysis could increase the confidence into the results.
 - o Proteomic analysis of the endocytic vesicle from a second cell type or a comparison with vesicles upon a different stimulation than growth factor signaling would also increase the confidence.
-
- Only very few experiments are confirmed in an additional cell type. Most experiments are conducted either in Swiss 3T3 fibroblasts or in MEFS. A general confirmation of the most important results in both cell types would be beneficial. In addition, a confirmation in a primary cell type such as hepatocytes would be intriguing.
-
- A stimulation with not only PDGF but also EGF or HGF in most experiments would strengthen the general validity of the concept.
-
- More evidence for mitochondrial co-localization of the endocytic vesicles would be required since hexokinases have been reported to localize to multiple compartments. PLA with a canonical mitochondrial outer membrane marker protein for example VDAC or mitofusin could show this.
-
- The electron microscopy only shows the accumulation of the nanobodies in the purified vesicles, but electron microscopy would be the ultimate method of choice to prove the accumulation of glycolytic enzymes in the vesicles via immunogold labeling or the interaction of the vesicles with the mitochondria.
-
- Figure legend for Figure4. requires more details. Does the hierarchical clustering show only the significantly altered or all measured metabolites? It is surprising that there are no metabolites which remain constant between both cell types. The p values seem not to be corrected for multiple hypotheses testing.
-
- The statement: "Source data for figures are available from the corresponding author upon request." Should not be acceptable for a paper containing proteomics and metabolomics data. All original data should be made publicly available with publication. Proteomics data should be uploaded on PRIDE.

Response to Reviewers

REVIEWER COMMENTS

Reviewer #1 (Remarks to the Author):

This manuscript explores glucose uptake in response to growth factor-stimulated endocytosis of the glucose transporter 1. The results present a novel mechanism in the regulation of glycolysis and will be of interest to a wide range of investigators studying glycolysis and/or receptor endocytosis. The experiments use appropriate controls and the data is clearly presented. The manuscript would benefit from additional rationale for experimental design, further quantification of existing data and additional experimental replicates (three independent experiments per assay). Further, the authors should test whether knockdown of PDGFRalpha and/or PDGFRbeta attenuates GLUT1 internalization and glucose uptake in response to PDGF-BB ligand treatment.

We appreciate Reviewer #1's comment that our research "will be of interest to a wide range of investigators studying glycolysis and/or receptor endocytosis." We have performed additional experiments to address his/her concerns and added some clarifying information to the revised manuscript. We hope that he/she finds these responses illuminating, and that they resolve his/her remaining concerns.

Major comments:

1. The authors do not state which PDGFR dimer they hypothesize is being activated in response to PDGF-BB ligand stimulation in their system. PDGF-BB stimulation of fibroblasts has previously been shown to result in formation of PDGFRalpha/beta heterodimers and PDGFRbeta homodimers. The authors frequently switch between assaying the two receptors - for example, the immunofluorescence assays in Figure 2 use antibodies against PDGFRalpha, while the PLA studies in the same figure use antibodies against PDGFRbeta. The authors need to clearly present their rationale for assaying one receptor over another in each experiment.

We thank the Reviewer for pointing this concern. We use goat polyclonal anti-PDGFR α antibodies (R&D AF1062) for most immunofluorescence staining and goat polyclonal anti-PDGFR β antibodies (Santa Cruz sc-1627) for some of immunofluorescence staining and PLA. We of course do not intend to cherry-pick data by switching anti-PDGFR antibodies, but rather because of their applicability to each assay. Importantly, as these receptors colocalize to the same endosomes upon stimulation with PDGF as shown in Supplementary Fig. 3a (former Extended Data Fig. 2a), we think we have observed the same structures and interactions using these two antibodies.

For immunofluorescence staining, we predominantly utilize the polyclonal anti-PDGFR α antibodies due to their brighter fluorescence intensity compared to the goat polyclonal anti-PDGFR β antibodies. While we did not explicitly specify the reason, the difference may be attributed to variations in affinities of antibodies or expression levels of PDGFRs.

On the other hand, we use the polyclonal anti-PDGFR β antibodies for PLA because PLA requires a pair of antibodies raised in different animals and these antibodies should recognize epitopes in the same cellular compartment. We utilized commercially available antibodies against cytoplasmic glycolytic enzymes (GAPDH, etc.) or the cytoplasmic tail of GLUT1 produced in rabbits, so the counterpart should be an antibody that labels the cytoplasmic surface of PDGFR-endocytic vesicles raised in a non-rabbit animal. Thus, we employed the goat polyclonal anti-PDGFR β antibodies because they bind to the cytoplasmic region of mouse PDGFR β and were previously validated for PLA (1), whereas our anti-PDGFR α antibodies bind to the *extracellular domain*, and we could not find other available anti-PDGFR α antibodies that fulfil all the requirements.

To prevent any confusion, we have included the statement “Hereafter, we targeted PDGFR β for PLA because of the antibody applicability.” in P.4 L.23 (L.114). We also added microscopic images that show colocalization of PDGFR β and GLUT1 upon PDGF-stimulation in Fig. 2a.

2. It is unclear from the images in Figure 3b if dynamin depletion inhibits PDGF-BB-dependent endocytosis of PDGFR α , as the magenta signal for the receptors has a similar expression pattern in control and TKO MEFs. The authors need to show single channel images for these experiments and quantify the extent of PDGFR α internalization in each case.

We added single channel images to Fig. 3c (former Fig.3b). Please note that immunofluorescence staining of PDGFR α of unstimulated cells shows almost entire-cell labelling because of thinness of fibroblasts and of optical limitation of confocal microscope for z-directional resolution, whereas concentrated vesicle structures (endocytic vesicles) appear in PDGF-stimulated cells. We also quantified surface PDGFR α of dynamin TKO fibroblasts and shown in Supplementary Fig. 6a in the revised version, which indicates that receptor endocytosis occurs in control MEFs but not in dynamin TKO MEFs.

3. The authors focus on the co-endocytosis of GLUT1 and the PDGFRs, and a subsequent increase in glucose uptake, in response to PDGF ligand stimulation. Though Figure 3h indicates that this phenomenon may also take place in response to a subset of additional growth factors, it will be critical to the authors' hypothesis that they demonstrate that knockdown of PDGFR α and/or PDGFR β attenuates GLUT1 internalization and glucose uptake in response to PDGF-BB ligand treatment.

We appreciate the Reviewer for suggesting this important point. We employed siRNAs against PDGFRs and assessed glucose uptake. As shown in Fig. 3b (and in Supplementary Fig. 5c, 5d) and described in P.5 L.18 (L.140) in the new version, single knockdown of either PDGFR did not interfere PDGF-evoked glucose uptake, whereas double-knockdown suppressed it. In addition, GLUT1 colocalized to endosomes with one of PDGFRs in cells where the other receptor was suppressed (Supplementary Fig. 5d). These data indicate redundant roles of PDGFRs in this process, and that stimulation-dependent glucose uptake as well as GLUT1 endocytosis can occur via homodimerization of either receptor. PDGFR α -PDGFR β heterodimer would also be involved in this process, given that PDGFR α and PDGFR β physically interact when assessed by immunoprecipitation in stimulated cells (please see below) and that PDGFR α and PDGFR β colocalize to the same endosomes upon ligand stimulation (Supplementary Fig. 3a).

Fig.1. Heterodimerization of PDGFR α and PDGFR β . Serum-starved Swiss 3T3 cells were stimulated with PDGF-BB (50 ng/ml) for 10 min. Immunoprecipitates and total cell lysate (input) were then subjected to immunoblotting with indicated antibodies

4. The figure legends often indicate that two independent experiments were performed, when 3 independent experiments would be more appropriate and strengthen the authors' conclusions. Further, in several instances legends indicate 3 technical replicates and 2 independent experiments. As there are only 3 datapoints on several of the relevant graphs, it is unclear what data was averaged for visualization. In instances where individual cells are analyzed (i.e. PLA), the authors are advised to use superplots to indicate individual datapoints as well as averages from independent experiments. Statistics should then be performed using the average values per experiment.

We greatly appreciate the Reviewer's instruction. In the new version, we visualize results of glucose uptake assays and other in vitro assays as superplots. These graphs show individual technical replicates (light-color dots), independent biological replicates (deep-color dots, averages of technical

replicates), and averages of biological replicates and their standard deviations (bars and error bars). Statistical analyses were performed using the values of biological replicates.

For quantification of immunofluorescence staining and image-based colocalization (Pearson's coefficients) analyses, however, we performed statistical analysis using datasets from one of the biological replicates (as technical replicates) to avoid batch effects among different experiments.

For PLA, we had originally designed the experiments assuming the individual cells are the subject of analyses, and performed 2 repeats of assays for each. Especially, although current recognition in the field is not necessarily simple, as mentioned "Further complicating the issue of sample number, what is considered as "an individual sample" depends on the experimental question." (2), we thought signals in individual cells, rather than averages, should be analyzed because of the large deviation of PLA signals in each cell, to answer our question whether there are interactions between PDGFR-vesicles and glycolytic enzymes. Importantly, absolute number of puncta (signals) of PLA, based on a commercially available kit from Sigma-Aldrich, is highly dependent on the batch of reagents in the kit (and probably on titers of primary antibodies and perhaps subtle differences in experimental conditions such as temperature). As shown below, numbers of puncta to detect the same interaction was substantially different between experiments performed with different batches of kit in different years, while still maintaining the magnitudes of changes evoked by PDGF stimulation.

Fig. 2. Batch effect of Duolink kits on PLA. Serum-starved Swiss 3T3 fibroblasts were stimulated or untreated for 10 min, and then subjected to PLA with indicated pairs of antibodies. Colors of dots in the graph represent independent biological repeats. Bars represent mean \pm SD of PLA signals per cell. P values were calculated using two-tailed Welch's t test.

Thus, we concluded that PLA is not an assay that one can directly compare the absolute number of signal puncta in different experiments (unless they are performed as groups in the same conditions as

our current sets of data), rather it is semi-quantitative in the same experiment. In the revised manuscript, we replaced the graphs for PLA with those of current dataset, but indicating replicates, because of the difficulty in incorporating additional data. We hope the Reviewer would agree that our data are still informative, which strongly support our model.

Minor comments:

1. Given the changing meaning of the label “control” throughout the manuscript, the label “PDGF-BB 0 min” might be more clear when comparing unstimulated versus ligand-stimulated cells.

We apologize for the confusing labels. Because they were not sampled at 0 min of PDGF treatment, rather at the same time as stimulated cells, we relabeled them as “-” or “unstimulated” in the revised version.

2. The authors should comment on the lack of plasma membrane PDGFR α signal in the absence of ligand stimulation – for example, in Figures 2a,b.

We thank the reviewer for this question. As mentioned above, fibroblasts used in this study are thin as a single confocal plane (1 AU) is enough to detect the whole cellular thickness in part due to the optical limitation of confocal microscope for z-directional resolution. For this reason, immunofluorescence staining of PDGFR α of unstimulated cells, which do not have PDGFR endocytic vesicles, shows almost entire-cell labelling, although one might expect signals alongside the cellular periphery for PM proteins. We added this explanation, “observed as entire cellular shapes because of the thinness of the cells” in P.4 L.7 (L.98) to avoid any confusions.

3. The authors should speculate why introduction of the control siRNA in extended data Figure 3c lowers PLA puncta per cell numbers compared to Figure 2c.

We speculate that this difference is due to usage of different batch of PLA kits in this experiment as discussed in the response to Major Comment #4. We added a note regarding this problem in P.16 L.9 (L.472) in the Methods section.

4. The authors should add single channel images for the experiments in Figure 2d.

We added single channel images to Fig. 2d.

5. The conclusion “Basal glucose uptake in TKO MEFs was slightly enhanced compared with that in control MEFs (Fig. 3c)...” is misleading, as there is a statistically significant difference.

We apologize for the confusing description. The word “slightly” was omitted in the revised version in P.6 L.6 (L.159).

6. On page 7, line 10, change “update” to “uptake”.

We apologize for the typo. We corrected it in the revised manuscript.

7. On page 7, line 24, change “3g” to “3h”.

We apologize for the mislabeling. We corrected it in the revised manuscript.

8. Antibody dilutions should be provided for all assays.

We thank the Reviewer's advice. We attached an antibody list with dilutions as Supplementary Table.

Reviewer #2 (Remarks to the Author):

Tsuzumi et al presents experiments demonstrating platelet derived growth factor (PDGF) induced stimulation of cells harboring PDGFR result in endocytosis of glucose transporter 1 (GLUT1) via the same endocytic vesicles. The author also present experiments suggesting evidence that critical glycolytic enzymes localize in regions of mitochondria in proximity to the endocytosed vesicles containing PDGFR/GLUT1. On the basis of the experiments presented in the manuscript the authors propose that “that PDGF stimulated glucose uptake depends on receptor/transporter endocytosis” providing evidence for “a new layer of regulation for glycolytic control governed by cellular membrane dynamics”.

Major comments:

1) The authors provide evidence for transport of GLUT1-containing vesicles to mitochondria by comparing the localization of endocytosed PDGFR, which colocalizes with endocytosed GLUT1 (Fig. 2a, b), and the hexokinases HK1 and HK2, which are known to localize to the cytoplasmic surface of mitochondria (Extended Data Fig. 2c). To complement these experiments, the authors perform a proximity ligation assay (PLA) using antibodies against PDGFRb and HK1 and HK2. The authors show that PLA signals, which indicate close proximity of PDGFR-containing vesicles to mitochondria, are

significantly increased after PDGF stimulation (Fig. 2c). However, this is not surprising since the number of endocytic vesicles containing PDGFR increases with PDGF stimulation and random encounters of these vesicles with HK1 and HK2 could produce a PLA signal. As a control experiment, the authors perform PLA after siRNA mediated HK1 and HK2 knockdown, but this merely validates that the assay works properly. How meaningful is the increase in PLA puncta per cell after PDGF stimulation? For example, does trafficking of unrelated endocytic vesicles (i.e., GLUT1-negative vesicles) also produce a PLA signal? It is important to show in some way that the PLA signal does not simply reflect an increase in the number of endocytic vesicles after PDGF stimulation.

We thank the Reviewer for pointing this important issue. While he/she kindly suggested GLUT1-negative unrelated endocytic vesicles, we faced difficulty detecting an interaction between such vesicles and mitochondria due to our lack of knowledge regarding the specific target protein(s) for these vesicles. So, we addressed this issue by treating cells with dynein inhibitors to assess whether or not PLA signals simply reflect an increase in the number of endocytic vesicles after PDGF stimulation.

As shown in Supplementary Fig. 7a-c, dynein inhibitors did not affect ligand-evoked PDGFR autophosphorylation and co-endocytosis with GLUT1. However, they did suppress PLA signals for stimulation-dependent PDGFR β -HK1 proximity (i.e. endocytic vesicle-mitochondria proximity). Thus, we concluded that increased PLA signal by PDGF does not simply reflect receptor endocytosis but requires intact intracellular vesicle transportation. Of note, we speculate that PDGFR endocytic vesicle-mitochondrial “collision” would occur alongside microtubules because both are transported by microtubule-associated dynein motors, and chances of such proximities would be “random” on microtubules. Meanwhile, stimulation-evoked increase in PLA signals for PDGFR β -GAPDH was not inhibited by dynein inhibitors. Given that it was strongly suppressed by dynamin TKO (Supplementary Fig.8), PLA signals for proximities between vesicles and cytoplasmic glycolytic enzymes may simply reflect increased number of endocytic vesicles.

Our observations thus indicate that some of stimulation-evoked interactions between PDGFR β and glycolytic enzymes could be mediated by random encounter, while the others, especially inter-organellar interactions, are not. We added a sentence “These dynein inhibitors suppressed PDGF-evoked PDGFR β -HK1 PLA signals without interfering stimulation-enhanced PDGFR phosphorylation, PDGFR/GLUT1 co-endocytosis, and PDGFR β -GAPDH proximity, also suggesting that enhanced PLA signals do not simply reflect an increase in the number of endocytic vesicles (Supplementary Figs. 7a-c).” in P.7 L.18 (L.202).

2) The authors show that limiting glucose in the presence of PDGF (without pyruvate and FBS) induces cell death over a 3-day period more so for dynamin TKO MEFs than for control MEFs (Fig. 4a). However, it is not clear how this relates to the regulation of cellular carbon metabolism by PDGF through an

endocytosis-dependent mechanism (versus a long-term transcription/translation-dependent mechanism) when PDGFR/GLUT1 trafficking is likely over after 1-hour PDGF stimulation because, as the authors clearly show, PDGFR is largely degraded by this time point (Extended Data Fig. 4d). How do the authors explain this?

We appreciate the Reviewer for giving an opportunity to clarify this important point. As the Reviewer mentioned, the survival assay and the metabolomic analysis in Fig. 5 (former Fig. 4) were designed to assess the significance of glucose uptake via endocytic vesicles in more continuous setting with lower concentration of PDGF than in other experiments in Figs. 2-4. We agree that detailed cellular condition in this setting is quite informative.

In the revised manuscript, we present data on 1) expression levels of glycolytic enzymes, 2) surface PDGFR α expression, and 3) intracellular localization of PDGFR α and GLUT1 under the same condition as our metabolomic analysis (0.1 g/L glucose in the presence of low PDGF for 24 h). As shown in Supplementary Fig. 9, a comparison between control and dynamin TKO fibroblasts revealed comparable total (and surface) expression levels of PDGFRs as well as glycolytic enzymes, excluding the possibility of long-term effect on these proteins. Importantly, probably because of continuous culture with low concentration of PDGF without serum-starvation, PDGFRs were still detectable. Furthermore, immunofluorescence staining showed vesicular structures containing both PDGFR α and GLUT1 in control MEFs.

These data strongly indicate that endocytosis-dependent glucose uptake continues to occur in this experimental condition and exclude at least direct transcriptional/translational suppression of the glycolysis pathway components by dynamin knockout. It is still possible that there is a yet unknown dynamin-dependent but endocytosis-independent survival mechanism that appears only when the cells sense glucose depletion, we consider it is simpler and more cogent to attribute it to endocytosis-dependent glucose uptake.

We included these data in Supplementary Fig. 9 and described in P.9 L.8 (L.254).

Minor comment:

It is not clear in either figure legends or Methods how confocal microscopy images are presented. Please indicate whether images are single confocal planes or maximum intensity projections.

We apologize for the lack of information. All microscopic images in figures are single confocal planes. As described in the answer to the Reviewer #1 minor point #2, please note that fibroblasts used in this study are thin that even a single confocal plane (1 AU) is enough to detect the whole cellular thickness. For this reason, immunofluorescence staining of PDGFR α of unstimulated cells results in almost entire-cell labelling. We stated that images are from single confocal planes in P.16 L.21 (L.484) and P.17 L.13 (L.505).

All in all, a revised manuscript following a major revision could be considered for publication.

We thank the Reviewer#2 for his/her comments on our manuscript. We hope that the additional results and information described above can bolster our conclusions.

Reviewer #3 (Remarks to the Author):

The authors describe an exciting novel mechanism that allows fibroblast cell lines to boost their glucose uptake and utilization upon growth factor stimulation. Using nanoparticle-based isolation of PDGFR-positive vesicles, the authors highlight an inducible mechanism for GLUT1-endocytosis that delivers the receptor directly to glycolytic enzymes. Although the manuscript provides compelling PLA data confirming co-localization of PDGFR-containing vesicles with the glycolytic machinery, some major points need to be addressed to strengthen the hypothesis.

We thank the Reviewer#3 for his/her positive comment that we “describe an exciting novel mechanism”. We hope that the additional results strengthen our model enough for publication.

Major comments:

1) The author use PDGF-mediated induction of GLUT1 endocytosis to derive all their observations. It is unclear why they used this growth factor in MEFs? Is this regulation of GLUT1-endocytosis a common feature for growth factors in fibroblast cells? What do other growth factors would do, like EGF? The old paper on EGF induced glucose uptake was never followed up. Would this mechanism be also true for other cell types and other growth factors? These important questions need to be addressed.

We use PDGF mainly because Swiss 3T3 fibroblasts (and other MEFs) express PDGFRs. Although not mentioned in the manuscript, some of the authors previously focused on PDGF-dependent endosomal production of reactive oxygen species (ROS) in fibroblasts (1) and had known that fibroblasts respond well to PDGF-BB. We therefore used PDGF and fibroblasts as a model for growth factor-dependent receptor endocytosis in this study.

Concerning other growth factors, we respectfully assert that we presented data for glucose uptake in MEFs stimulated with EGF, bFGF, HGF, or insulin in Fig. 4c (former Fig. 3h). In this experiment, we confirmed that at least bFGF and HGF induce glucose uptake in a dynamin-dependent manner, suggesting that receptor endocytosis-dependent glucose uptake also occurs in cells stimulated with these growth factors. Although we did not observe a statistical significance, EGF also showed a trend

enhancing glucose uptake depending on dynamin. To avoid any confusion, we added description “We also observed a trend of increase in glucose uptake in EGF-treated MEFs as previously reported (Fig. 4c)” in P.8 L.7 (L.217).

Concerning other cell type and growth factors, we assessed HeLa cells which is known to respond to EGF. As shown in Supplementary Fig. 7f, HeLa cells showed EGF-evoked glucose uptake depending on GLUT1 and cytoplasmic dynein. We added description in P.8 L.12 (L.227).

Another example is NB1 neuroblastoma cells, which overexpress another RTK, ALK, as shown below. Treating NB1 cells with a specific ALK inhibitor crizotinib for 10 min in the presence of fetal bovine serum (FBS) resulted in decreased glucose uptake. In addition, Glucose uptake was suppressed by treating cells with dynein inhibitors, ciliobrevin D or dynarrestin. These suggest that the mechanism could be applicable to RTK-overexpressing cancer cells as well, although this data is not included in the manuscript because we think it is preliminary and beyond the scope of this study.

Fig.3. Glucose uptake in NB1 neuroblastoma cells. NB1 neuroblastoma cells were treated with inhibitors or vehicle as indicated for 10 min in the presence of FBS and were then subjected to glucose uptake assay in the presence of 2DG with vehicle or the inhibitors. Graph shows 2DG6P normalized to total cellular proteins. Bars and error bars in the graph show means of 3 technical replicates and \pm SD from one of 2 biological replicates. P values were calculated using ANOVA and post-hoc Tukey's tests.

2) The authors use high amounts of glucose (4.5 g/L) in all of their studies. In a physiological setting that would refer to a highly diabetic condition. Also, in conditions of tumor growth this concentration is way too high to be physiological. I acknowledges that these are the commonly used cell culture conditions for such experiments, however, I question how physiological the responses really are? Does this hold true also for lower glucose concentrations, which would reflect more 5 mM of an in vivo situation?

We thank the Reviewer for providing important point. However, since cells were subjected to glucose uptake assay in 1 mM 2DG in the absence of glucose in this study, as described in the Method section, the experiment is presumably not directly influenced by the high concentration of glucose (4.5 g/L, 25 mM). To assess whether long-term culture in different glucose concentration affects PDGF-dependent glucose uptake, we cultured control and dynamin TKO MEFs in low glucose (1 g/L, 5.6 mM) DMEM (nacalai, #08456-65) for 5 days and then subjected them to glucose uptake assay. The result, shown in Supplementary Fig. 6d, indicated that cells cultured in low glucose medium still showed PDGF-evoked glucose uptake depending on dynamin, similarly to the cell cultured in high glucose medium. We added description on P.6 L.16 (L.169).

3) How are the glycolytic proteins associated with the endocytic vesicles. Via tethers? Anything from the interactome?

We thank the Reviewer for pointing the important question. However, the mechanism has been left unknown yet. We hope that the Reviewer would agree that clarification of the molecular mechanism of endocytic vesicle-glycolytic enzyme interaction is beyond the scope of this manuscript.

As far as we tried, the complex of PDGFR and glycolytic enzymes was not detectable at least when assessed by immunoprecipitation of the receptors in the presence of detergent (data not shown). Similarly, we have not found candidate scaffold protein(s) in STRING database (<https://string-db.org/>), perhaps because most of the interactions in the database would be assessed using unstimulated cells in the normal immunoprecipitation condition in the presence of detergent. Additionally, detection of direct interaction between glycolytic enzymes and phospholipid by employing PIP Strips™, membranes with phospholipid array, is unsuccessful so far (data not shown). There reported some candidate scaffold proteins including SLC4A1 (Band 3), which targets glycolytic enzymes to the plasma membrane in erythrocytes (3). Although SLC4A1 suppresses glycolysis and was not in our endocytic vesicle protein list, other scaffold may be involved in recruitment of glycolytic enzymes to the vesicles.

We added description “The mechanism for formation of glycolytic enzyme complex adjacent to endocytic vesicles remains to be elucidated. There may be scaffold proteins like Band 3 (SLC4A1), which targets glycolytic enzymes to the plasma membrane in erythrocytes (36), although it suppresses glycolysis and is not in our list of endocytic vesicle proteins.” in P.11 L.1 (L.309).

4) Line 12 on page 4: The authors state that localization of HK1/2 to the mitochondrial outer membrane is shown in Ext. Fig. 2c, 3a, and 3b, although the referenced data does not show this. This needs to be addressed. Co-stainings and quantifications with mitochondrial markers, like Tom20 need to be conducted.

We thank for the Reviewer's remark. We show co-staining of the fibroblasts with anti-PDGFR α , anti-HK1 or HK2, and anti-TOMM20 antibodies in Supplementary Fig. 3c and described in P.4 L.15 (L.106) to indicate localization of HK1/2 to the mitochondria.

5) Line 15 on page 4: Where did the authors observe proximity of PDGFR α -containing vesicles to mitochondria? This data is not included in the manuscript and needs to be presented.

We apologize for confusing description. At this explanation, we just speculated proximity of PDGFR α -containing vesicles to mitochondria, and therefore employed PLA to assess this possibility. To avoid confusions, we changed the expression to "we observed that *some* PDGFR α -containing vesicles *seemingly* localized in proximity to mitochondria" in P.4 L.18 (L.109).

6) The authors should discuss their data in a broader fashion. There are studies related to dynamin 2 KO in b-cells and other metabolically active organs, which show connections to glucose uptake and insulin secretion, thus supporting this proposed connection. In addition, there are plenty of studies on endosomal regulators in mice, which reveal an impact in glucose uptake, glycolysis and glucose metabolism in general which should be incorporated into the bigger picture of the observation presented here and needs to be thought about and discussed.

We appreciate the Reviewer for this advice. We added 3 references in Discussion, "In particular, previous studies suggest decreases in glucose uptake in dynamin-deficient pancreatic β cells and T lymphocytes (38-40). Although the primary mechanisms identified were deficient insulin secretion in β cells and decreased GLUT1 expression in T lymphocytes, receptor endocytosis-dependent glucose uptake may be involved in these processes in part." in P.11 L12 (L.320).

7) The authors propose increased GLUT1-mediated glucose transport to cause increased intracellular glucose accumulation by ruling out enhanced glucose-retention by testing hexokinase activity. However, as GLUT1 mediates glucose transport via facilitated diffusion, the authors should assess glucose levels within their isolated vesicles. This is especially in light of increased glucose concentrations nearby vesicles, as assessed by Green Glifon4000 (Fig. 3g).

We appreciate the Reviewer for this suggestion. To assess if there is actually glucose in the

endocytic vesicles, we directly quantified glucose in PDGFR-containing endocytic vesicle fraction. As result, we detected glucose in the fraction only when the entire procedure was performed in the presence of GLUT1 inhibitor BAY876. This data leads to important conclusions, that isolated endocytic vesicles contain glucose, and that the glucose molecules can defuse out from the vesicles in a GLUT1-dependent manner, both of which agree with our model. The data is shown in Supplementary Fig. 7d and described in P.8 L.2 (L.217). As we cannot determine the exact number and sizes of endocytic vesicles in the fraction, estimating the absolute levels of glucose in the vesicles is challenging. Nevertheless, we hope the Reviewer agrees that we have, at least, demonstrated the presence of glucose in PDGFR-containing endocytic vesicles.

8) How to explain unchanged surface levels of GLUT1 in MEFs, given the increased endocytosis of GLUT1 together with PDGFR as shown upon PDGF treatment? In 3T3s, there is a decrease (Ext. Data Fig. 4c). Any effect on recycling of GLUT1?

We appreciate the Reviewer for this question. As he/she mentioned, we did not detect significant decrease of surface GLUT1 in control MEFs in Supplementary Fig. 6c (former Extended Data Fig. 4f), whereas surface GLUT1 was decreased in Swiss 3T3 cells with statistical significance in Supplementary Fig. 5f (former Extended Data Fig. 4c). However, as one could see, the PDGF-dependent decrease in surface GLUT1 was not substantial even in Swiss 3T3 MEFs (approximately -11%). We therefore suspect that the magnitude of decrease in surface GLUT1 in control MEFs was not strong enough, even if there was. It is noteworthy, however, we confirmed that PDGFR α and GLUT1 are colocalized to endocytic vesicles in stimulated control MEFs by immunofluorescence staining (Fig. 3c), indicating that the co-endocytosis occurs in these cells. Additionally, the most important point of this result is that PDGF-dependent enhanced glucose uptake is not due to *increase* in surface GLUT1.

We think the reason why the degree of decrease of surface GLUT1 upon PDGF-stimulation is less than that of surface PDGFR is simply due to excess GLUT1 molecules when compared to PDGFRs, rather than recycling within this short period of time. We describe this relatively unchanged surface GLUT1 levels in P.5 L.23 (L.145).

9) Please adjust the order of figures with their respective labeling according to the order in which the data is referenced in the results section to allow an easier flow of reading.

We apologize for inconvenience. We modified the order of figures in revised manuscript.

Minor comments:

- 1) Line 24 on page 4: TPI should read TPI1.
- 2) Line 4 on page 6: Please correct "Fig, 3d" to "Fig. 3d".

3) Line 26 on page 6: “making it is unlikely” should read “making it unlikely”.

We apologize for these typos and mislabeling. We corrected them in the revised manuscript.

4) Lines 11-14 on page 7: This sentence is confusing and should please be revisited.

According to the reviewer’s comment, we simplified the description in P.7, L.22 (L.206).

5) Lines 19-21 on page 7: Please also show Green Glifon4000 co-staining with PDGFR α/β in cells not treated with PDGF.

We added images for unstimulated cells in Fig. 4b in the revised manuscript.

6) Line 22 on page 7: “(bFGF)- and“ should read “(bFGF) and”.

7) Line 24 on page 7: Referenced Fig. 3g should instead reference to Fig. 3h.

We apologize for these typos and mislabeling. We corrected them in the revised manuscript.

8) Lines 2-4 on page 9: This conclusion is questionable, as in addition to increased AMP levels, ADP is also elevated.

We appreciate the Reviewer’s comment. We omitted this sentence.

References

1. R. Tsutsumi et al., Assay to visualize specific protein oxidation reveals spatio-temporal regulation of SHP2. Nature Communications 8, 466 (2017).
2. R. A. Senft et al., A biologist’s guide to planning and performing quantitative bioimaging experiments. PLOS Biology 21, e3002167 (2023).
3. H. Chu, P. S. Low, Mapping of glycolytic enzyme-binding sites on human erythrocyte band 3. Biochemical Journal 400, 143–151 (2006).

REVIEWER #4

In their study "Endocytic Vesicles as Conduits for Glucose Uptake Triggered by Growth Factor Stimulation," Tsutsumi et al. describe a novel pathway for glucose uptake in response to growth factor signaling. Employing proteomics of endocytic vesicles, different imaging techniques, metabolite quantification, and metabolic assays, the authors state that upon activation by growth factors, cells initiate the formation of endocytic vesicles enriched with the glucose transporter Glut1, key glycolysis enzymes, and glucose itself. These specialized vesicles appear to be selectively transported into close proximity with the mitochondria. They provide data that, upon perturbation of the endocytic machinery cellular carbon metabolism is impaired.

While the finding is novel and deepens our understanding of cellular metabolism and the regulation of glycolysis, with profound implications for physiological processes, the study does bear some severe technical concerns regarding the proteomics data and certain experimental aspects. The proteomics analyses indicate that the enzymes might be enriched but it cannot be excluded that the detection is due to contaminations. Therefore, alternative approaches such as immunogold labeling electron microscopy would be required to give an ultimate prove of the concept. Furthermore, to reinforce the general applicability of this discovery, the study could benefit from the use of multiple cell lines for the main experiments or the use of alternative stimuli in addition to PDGF stimulation. This would bolster the robustness of their findings.

We thank the reviewer's comment that "the finding is novel and deepens our understanding of cellular metabolism and the regulation of glycolysis, with profound implications for physiological processes..." We hope our response below can clarify his/her concerns. *However, please note that we did not have time for additional experiments and for further revision of the manuscript because we received the Reviewer #4's comments from the editor after we finished the requested revision. We hope the reviewer would generously understand this unusual situation.*

While the discovery of an endocytosis mediated glucose uptake is in general interesting, some conceptual questions remain. How should the HK1 and HK2 residing at the outer mitochondrial membrane access the glucose transported in the vesicles? How should the glycolytic machinery be incorporated specifically into the vesicles? For this some interaction with receptors or a sorting mechanism would be required.

We respectfully clarify that the reviewer might have misunderstood our model. Our proposition does not involve the incorporation of glycolytic enzymes into the vesicles. Instead, we propose that

glucose within the vesicles can be released into the cytoplasm near the mitochondria through GLUT1. This released glucose can then undergo HK-dependent phosphorylation and subsequent glycolysis catalyzed by the cytoplasmic glycolytic enzymes.

Main points:

- While in the proteomics section, endocytic vesicles seem to be enriched, the high number of identified proteins indicates strong contaminations, what makes the results difficult to interpret. Moreover, the authors state that glycolytic proteins are enriched in the purified endocytic vesicles, but no enrichments towards other fractions of the purification process are shown, only peptide counts of the vesicle fraction. No general quality assessment of the proteomics data is performed. Only peptide counts for selected proteins are picked and shown as changed without any statistical testing or comprehensive data analysis. The following experiments would strongly increase the robustness of the proteomics experiments:

The Reviewer is correct that a large number of proteins was identified. However, when looking at the intensity of the proteins using the spectral counts as a quantitative metric (Supplementary Table 1), it is clear that there are only a few proteins that are dominating the sample and the other proteins are contaminants that we see with any prep due to the sensitivity of the mass spectrometer.

We also acknowledge that our MS data lacks other fractions than endocytic vesicles and replicates for statistics. This limitation stems from primarily due to resource limitations of the first author. However, spectral counts are an accepted method for label free protein quantitation (1). The reviewer is correct that replicates would make the case certainly stronger, fortunately the enrichment here is striking and in addition verified by immunoblotting (Fig. 1e) for most of the highlighted targets. Thus, while not ideal, the clear enrichment of the discussed proteins in the conjugated over unconjugated particles is strong.

Importantly, anyways, we think our conclusions in this study are not solely based on the MS result but rather on series of experiments including immunoblotting, immunofluorescence staining, PLA, glucose uptake assay, as well as in vitro assays. We hope the reviewer would agree that the MS result was merely a starting point of this study.

o Show the enrichment of glycolytic enzymes and GLUT1 in the endocytic fraction versus the input and the flow through. The enrichment profiles of enzymes for, for example fructose metabolism, purine metabolism, other transporters or signaling complexes such as MTOR should be shown as negative controls.

Please see below for immunoblotting of glycolytic enzymes in input (from the same blots of Fig. 1g). We apologize for the very weak signals in the 0.5% input lane. While we cannot specify exact enrichment ratios, some abundant enzymes (GAPDH, ENO1, etc.) exhibited relatively lower degrees of enrichment compared to PDGFRs. Unfortunately, we lack data for the flow-through fraction. However, considering the abundance of glycolytic enzymes, the mostly unchanged subcellular localizations of glycolytic enzymes upon PDGF-stimulation, and the potential limitations in the recovery of our endocytic vesicle enrichment, the decrease in glycolytic enzymes in the flow-through fraction compared to the input would be limited.

Fig. Enrichment of PDGFRs and glycolytic enzymes in the endocytic vesicle fraction. Total cell lysate (0.5% input) and indicated fractions were subjected to immunoblotting with indicated antibodies (the same experiment as Fig. 1g).

We thank for the reviewer's suggestion for negative controls. However, we believe that readers can easily access the enrichment profiles in Supplementary Table 1. Notably, for fructose metabolism, fructokinase and aldolase B were not detected in the endocytic vesicle fraction. In purine metabolism, ATIC and XDH were detected but other major enzymes AMPD, PPAT, ASL, HGPRT, APRT were not detected or not enriched. Regarding other SLC family membrane transporters, only a limited subset (31 out of more than 400 members) including Slc25A5, 25A3, 25A24, 25A4, 12A2, 25A12, 25A13, 3A2, 1A5, 16A1, 25A11, 29A1, 39A7, 38A2, 38A4, 25A10, 7A5, 25A20, 12A4, 12A7, 20A2, 27A4, 35A1, 35E1,

38A10, 4A7, 1A4, 6A6, 9A1, 25A1, and 4A2 were enriched in the fraction. The mTOR protein was also enriched. Anyways, we would emphasize the difficulty in specifying pathways that are NOT involved in the fraction but are present in the cells based on the current dataset.

o Purity could be increased by a density gradient or differential centrifugation before the magnetic bead-based enrichment, alternatively a second alternative purification method based on different separation techniques prior to proteomics analysis could increase the confidence into the results.

We totally agree that additional step(s) will improve the purity of the fraction. Once again, however, we think the initial MS analysis in this study merely raised the possibility of the presence of glycolytic enzymes in the fraction, prompting us to conduct a series of experiments to support our final model. While we recognize that our current protocol may lack sophistication, we view further refinement of the method as a potential major project in the future. It is also noteworthy that the simplicity and quickness of our current protocol may contribute to our new finding, given the rapid dynamics of RTK signaling. Furthermore, as discussed in the manuscript, although our data would indicate contamination of other organelle than endocytic vesicles, the current method potentially enables the detection of inter-organelle interactions, which might not be detectable in highly purified vesicles.

o Proteomic analysis of the endocytic vesicle from a second cell type or a comparison with vesicles upon a different stimulation than growth factor signaling would also increase the confidence.

Although we understand the reviewer's comment that comparisons to samples from totally different contexts are beneficial, we do not have data at this point. Especially, we hope he/she would agree that such proteomic analyses focusing on other cells (which may necessarily need other ligands than PDGF) or other stimulations (possibly GPCRs, LDLRs, TNFRs, TGFRs, TfR, etc.) are challenging per se, which should be pursued as independent projects, and would be beyond the scope of the current manuscript.

- Only very few experiments are confirmed in an additional cell type. Most experiments are conducted either in Swiss 3T3 fibroblasts or in MEFS. A general confirmation of the most important results in both cell types would be beneficial. In addition, a confirmation in a primary cell type such as hepatocytes would be intriguing.

We respectfully argue that we think the most important results including PDGF-induced co-endocytosis of PDGFR and GLUT1 (Figs. 2a, 3c), PDGF-evoked proximity between PDGFR and glycolytic enzymes (Figs. 2c, 2d, 2e, and Supplementary Fig. 8), and PDGF-stimulated GLUT1-dependent cellular glucose uptake (Figs. 3a, 3d), were confirmed in both Swiss 3T3 fibroblasts and MEFs. While primary hepatocytes were not included in our dataset, we expanded our analysis to include HeLa cells stimulated with EGF (Supplementary Fig. 7f), demonstrating the generalizability of our observations to some extent in the revised manuscript.

- A stimulation with not only PDGF but also EGF or HGF in most experiments would strengthen the general validity of the concept.

We acknowledge the Reviewer's suggestion that repeating most experiments with additional growth factors like EGF or HGF would be ideal. However, we sincerely hope the Reviewer would understand that the entire volume of data will double or triple if additional growth factors are tested in most experiments and that it is challenging because of our limited resources. We have already shown, at least, that bFGF and HGF are able to evoke dynamin-dependent glucose uptake in fibroblasts (Fig. 4c).

- More evidence for mitochondrial co-localization of the endocytic vesicles would be required since hexokinases have been reported to localize to multiple compartments. PLA with a canonical mitochondrial outer membrane marker protein for example VDAC or mitofusin could show this.

In the revised manuscript, we present evidence of exclusive colocalization of HK1 and HK2 with the mitochondria marker TOMM20 in fibroblasts (Supplementary Fig. 3c). We think this result, albeit indirect, supports the notion that PLA signals between PDGFR and HKs are indicative of the proximity between vesicles and mitochondria.

- The electron microscopy only shows the accumulation of the nanobodies in the purified vesicles, but electron microscopy would be the ultimate method of choice to prove the accumulation of glycolytic enzymes in the vesicles via immunogold labeling or the interaction of the vesicles with the mitochondria.

We apologize for repeating, but we do not claim that glycolytic enzymes are in the vesicles, but they present in the cytoplasm close proximity to the vesicles. Given that the entire localizations of most of glycolytic enzymes are not changed by PDGF stimulation (Supplementary Fig. 4a), immunogold

labeling of them probably do not show predominant accumulation because portions of these enzymes are likely in proximity to the vesicles.

Regarding the interaction between vesicles and mitochondria, we acknowledge the importance of elucidating the microscopic structure of this phenomenon. However, we hope the Reviewer agree that the current manuscript focuses on the roles of RTK endocytosis, GLUT1 and vesicle transportation in GF-evoked glycolysis, and that further detailed elucidation of the vesicle-mitochondria interaction is one of the important questions to be solved in the future by a subsequent project.

- Figure legend for Figure4. requires more details. Does the hierarchical clustering show only the significantly altered or all measured metabolites? It is surprising that there are no metabolites which remain constant between both cell types. The p values seem not to be corrected for multiple hypotheses testing.

The heatmap includes all 92 metabolites detected out of the 116 tested, representing standardized amounts for each metabolite. Absolute amounts of 92 metabolites can be found in Supplementary Table 2. The p values were calculated by Welch's t-tests as described in the Methods and in the Figure Legend. q values, when calculated using multiple t tests by False Discovery Rate (FDR) approach by Two-stage linear step-up procedure of Benjamini, Krieger and Yekutieli, with $Q = 5\%$, can be found in attached "Supplementary Table 2_2. .xlsx."

- The statement: "Source data for figures are available from the corresponding author upon request." Should not be acceptable for a paper containing proteomics and metabolomics data. All original data should be made publicly available with publication. Proteomics data should be uploaded on PRIDE.

As mentioned in the Methods, the mass spectrometric raw files for the endocytic vesicle fraction are accessible at <https://massive.ucsd.edu> and at www.proteomexchange.org., both widely recognized platforms for proteomics data deposition.

The metabolomic analysis data was obtained from a service provider HMT, but the MS raw data were not provided due to the inclusion of the company's confidential analytical parameters. All the absolute amounts of metabolites calculated are shown as Supplementary Table 2.

Reference

1. Choi M, et al. ABRF Proteome Informatics Research Group (iPRG) 2015 Study: Detection of

Differentially Abundant Proteins in Label-Free Quantitative LC-MS/MS Experiments. *J Proteome Res* 16, 945-957 (2017).

REVIEWERS' COMMENTS

Reviewer #1 (Remarks to the Author):

The authors have addressed my initial concerns. However, after reading the comments of Reviewer #4, I strongly agree that all proteomics data should be submitted to the PRIDE database and made publicly available upon publication.

Reviewer #2 (Remarks to the Author):

The authors addressed, for the most part, my comments to the original version of the manuscript and the revised manuscript is acceptable for publication in Nature Communication.

Reviewer #3 (Remarks to the Author):

The authors have addressed all comments thoroughly, leading to a significantly improved story.

Reviewer #4 (Remarks to the Author):

In their rebuttal letter, the authors have addressed most of the concerns raised during the initial review process. Their arguments are convincing and have clarified several points that were previously misunderstood. Specifically, clarification regarding the model's interpretation that glycolytic enzymes are within the vesicles is appreciated, although it raises the possibility that other readers might also misinterpret this aspect. To avoid potential misunderstandings, it would be beneficial for the authors to further clarify their model in the manuscript on how glycolytic enzymes interact with vesicles. Including a discussion of existing studies that explore complex formations or interactions between glycolytic enzymes and membrane proteins would help to understand the model. The main limitation of this study is the absence of a primary cell model to validate the biological relevance of the findings. Since the authors did not have the opportunity to address this in the previous revision, acknowledging this limitation, and discussing that the findings are restricted to immortalized cell lines would be beneficial.

Response to Reviewers

REVIEWER COMMENTS

Reviewer #1 (Remarks to the Author):

The authors have addressed my initial concerns. However, after reading the comments of Reviewer #4, I strongly agree that all proteomics data should be submitted to the PRIDE database and made publicly available upon publication.

We appreciate Reviewer #1 for commenting that we have addressed his/her concerns. Regarding MS data for proteomic analysis, raw files are deposited in public databases as described in "Data Availability" section (P.21 L.8).

Reviewer #2 (Remarks to the Author):

The authors addressed, for the most part, my comments to the original version of the manuscript and the revised manuscript is acceptable for publication in Nature Communication.

We appreciate Reviewer #2 his/her comment that our manuscript is acceptable for publication.

Reviewer #3 (Remarks to the Author):

The authors have addressed all comments thoroughly, leading to a significantly improved story.

We appreciate Reviewer #3 for giving the opportunity to improve our manuscript and for his/her comment that we have addressed his/her concerns thoroughly.

Reviewer #4 (Remarks to the Author):

In their rebuttal letter, the authors have addressed most of the concerns raised during the initial review process. Their arguments are convincing and have clarified several points that were previously misunderstood. Specifically, clarification regarding the model's interpretation that glycolytic enzymes are within the vesicles is appreciated, although it raises the possibility that other readers might also misinterpret this aspect. To avoid potential misunderstandings, it would be beneficial for the authors to further clarify their model in the manuscript on how glycolytic enzymes interact with vesicles. Including a discussion of existing studies that explore complex formations or interactions between glycolytic enzymes and membrane proteins would help to understand the model. The main limitation of this study is the absence of a primary cell model to validate the biological relevance of the findings. Since the authors did

not have the opportunity to address this in the previous revision, acknowledging this limitation, and discussing that the findings are restricted to immortalized cell lines would be beneficial.

We appreciate Reviewer #4 for his/her comment that our responses are convincing. According to the Reviewer's advice, we added "Our data suggest that glucose in the endocytic vesicles is released into the cytoplasm near the mitochondria through co-endocytosed vesicular GLUT1." in Discussion to avoid potential misunderstanding (P.10 L.14). Additionally, the manuscript contains a discussion of existing studies that explore complex formations or interactions between glycolytic enzymes and membrane proteins (P.11 L.3). We also acknowledge the limitation of the current study by adding "further studies are required to clarify the involvement of receptor endocytosis in the control of cellular carbon metabolism, including its generality in vivo." (P.11 L.17).